# Recruitment of clathrin to intracellular membranes is sufficient for vesicle formation

**Cansu Küey, Méghane Sittewelle, Gabrielle Larocque[†], Miguel Hernández-González[†], Stephen J Royle***

Centre for Mechanochemical Cell Biology and Division of Biomedical Cell Biology, Warwick Medical School, University of Warwick, Coventry, United Kingdom

**Abstract** The formation of a clathrin-coated vesicle (CCV) is a major membrane remodeling process that is crucial for membrane traffic in cells. Besides clathrin, these vesicles contain at least 100 different proteins although it is unclear how many are essential for the formation of the vesicle. Here, we show that intracellular clathrin-coated formation can be induced in living cells using minimal machinery and that it can be achieved on various membranes, including the mitochondrial outer membrane. Chemical heterodimerization was used to inducibly attach a clathrin-binding fragment 'hook' to an 'anchor' protein targeted to a specific membrane. Endogenous clathrin assembled to form coated pits on the mitochondria, termed MitoPits, within seconds of induction. MitoPits are double-membraned invaginations that form preferentially on high curvature regions of the mitochondrion. Upon induction, all stages of CCV formation – initiation, invagination, and even fission – were faithfully reconstituted. We found no evidence for the functional involvement of accessory proteins in this process. In addition, fission of MitoPit-derived vesicles was independent of known scission factors including dynamins and dynamin-related protein 1 (Drp1), suggesting that the clathrin cage generates sufficient force to bud intracellular vesicles. Our results suggest that, following its recruitment, clathrin is sufficient for intracellular CCV formation.

**\*For correspondence:**
s.j.royle@warwick.ac.uk

**Present address:** [†]The Francis Crick Institute, London, United Kingdom

**Competing interest:** The authors declare that no competing interests exist.

## Editor's evaluation

This paper reports a striking finding, which should be of interest to cell biologists and biophysicists. The authors use an innovative approach to recruit clathrin to mitochondrial membranes, and observe the budding and fission of clathrin-coated vesicles. The study leads to a much clearer view of how the clathrin lattice functions in endocytosis.

## Introduction

Clathrin-coated vesicles (CCVs) are major carriers for cargo transport in cells (***Kaksonen and Roux, 2018***; ***Chen and Schmid, 2020***). CCVs carry cargo in three pathways: plasma membrane to endosome, from endosome to endosome, and between endosomes and *trans*-Golgi network (TGN). Due to experimental accessibility, CCV formation during endocytosis at the plasma membrane has been studied extensively. To what extent this event is representative of intracellular CCV formation is an open question.

We have known since the mid-1990s that there are four core components for CCV formation in endocytosis: cargo, adaptor, clathrin, and dynamin (***Robinson, 1994***). Clathrin forms the cage but cannot detect cargo nor membrane. An adaptor – the heterotetrameric AP-2 complex – recognizes cargo and membrane, and this recognition allows clathrin to bind to the adaptor and for pit formation

to begin (*Kelly et al., 2014*). The pit invaginates accompanied by clathrin polymerization, assisted by the adaptor itself (*Smith et al., 2021*). Eventually, through the action of the large GTPase dynamin, the vesicle is pinched off from the plasma membrane (*Damke et al., 1994*; *Antonny et al., 2016*). In the intervening years, a number of other proteins were identified that can be recruited to the forming clathrin-coated pit (CCP) (*Kaksonen and Roux, 2018*). Their structures were determined, their cellular dynamics analyzed and complex network diagrams were built (*Traub, 2011*). However, it is unclear how important many of these accessory proteins are to the CCV formation process. Which of these proteins are *mediators*, essential for general CCV formation, and which are *modulators*, whose activity may fine-tune the process or only be required for a minority of events? An example is proteins that induce membrane curvature, what is their contribution relative to clathrin polymerization in driving CCV formation (*Stachowiak et al., 2013*; *Sochacki and Taraska, 2019*)? These questions apply to CME but also to intracellular CCV formation, which has its own network of molecular players.

It has been assumed that anything we learn about CCV formation in endocytosis can be translated to intracellular CCV formation. Whilst there are many parallels, important differences in the core machinery are already apparent. The recognition of membrane by AP-2 is via PI(4,5)P$_2$ whereas for AP-1, the heterotetrameric adaptor complex for CCV formation at endosomes, this step is governed by PI4P and the GTPase Arf1 (*Ren et al., 2013*). In addition, imaging studies have questioned the requirement for dynamin in the fission of intracellular CCVs (*Kural et al., 2012*). In order to address these questions, we have developed a method to reconstitute CCV formation in living cells, from minimal components (*Wood et al., 2017*; *Smith et al., 2021*). Briefly, a clathrin-binding 'hook' is recruited to an 'anchor' at the plasma membrane using chemical heterodimerization. Because the initial cargo selection and membrane recognition steps are bypassed, we termed this method 'hot-wiring'. Using this method to trigger endocytosis at the plasma membrane led to de novo CCV formation (*Wood et al., 2017*). However because the plasma membrane is the site of endogenous endocytosis, it was difficult to (1) assess the role of endogenous accessory proteins in the induced events and (2) delineate induced pits from endogenous CCPs (*Wood et al., 2017*). To overcome these difficulties and to study the mechanism of intracellular CCV formation, we set out to reconstitute intracellular CCV formation on-demand, using the same principle. We show that CCPs can be induced on intracellular membranes, including the mitochondrial outer membrane. All stages of CCV formation are recapitulated at mitochondria, including fission; which appears to occur without the action of a scission molecule. Our data argue that the clathrin cage, in the absence of other additional factors, is sufficient to generate CCVs on intracellular membranes; highlighting a fundamental difference between CME and formation of intracellular clathrin carriers. Our findings also suggest that CCV formation after initial clathrin recruitment can proceed without the plethora accessory proteins that have been described.

## Results

### Inducing CCP formation on intracellular membranes

Previously, we described a method for inducing clathrin-mediated endocytosis at the plasma membrane (*Wood et al., 2017*; *Figure 1A*). We reasoned that by changing the membrane targeting anchor, it may be possible to form CCPs on intracellular membranes. Four distinct compartments were tested: mitochondria, endoplasmic retirculum (ER), Golgi, and lysosomes.

For targeting mitochondria, a mitochondrial anchor termed 'MitoTrap' was used which has the transmembrane domain of Tom70p, a mitochondrial outer membrane protein, fused to mCherry-FRB (*Robinson et al., 2010*; *Cheeseman et al., 2013*). For the clathrin hook, FKBP-β2-GFP was used which is composed of the hinge and appendage domains of the β2 subunit of AP-2 and an FKBP domain for inducible heterodimerization with FRB (*Figure 1A*). The clathrin hook was mostly cytoplasmic, upon rapamycin addition however, FKBP-β2-GFP became localized to the mitochondria within seconds and then small spots containing hook and anchor began to form (*Figure 1B* and *Figure 1—video 1*). A clathrin-binding deficient β2 hook (FKBP-β2 Y815A/ΔCBM-GFP) and a control hook, GFP-FKBP, showed mitochondrial localization but no spot formation following rapamycin addition, suggesting that the spots that form are CCPs (*Figure 1C*). We therefore termed these spots 'MitoPits'.

According to our model, the MitoPits should be spatially organized as follows: mitochondrial matrix followed by mitochondrial membrane and anchor, then the clathrin hook. To test this, line profiles

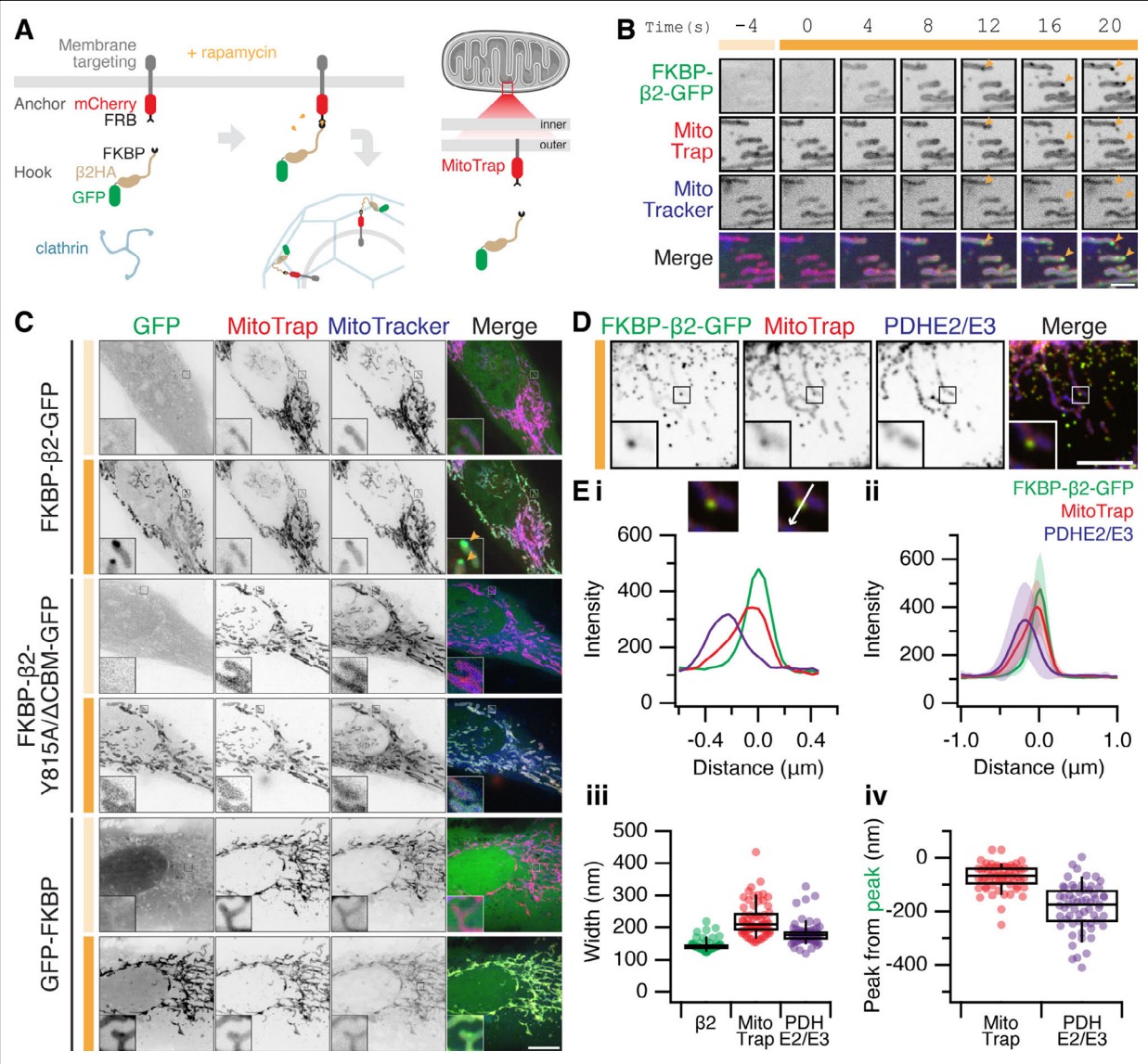

**Figure 1.** The formation of MitoPits. (**A**) Schematic representation of clathrin-coated pit induction. The system, triggered by rapamycin, consists of a membrane anchor (mCherry-FRB fused to a membrane targeting domain) and a clathrin hook (clathrin-binding protein fused to FKBP and GFP). Targeting the anchor to mitochondria using Tom70p in MitoTrap. (**B**) Stills from live cell imaging of a HeLa cell expressing FKBP-β2-GFP (green), and the anchor, MitoTrap (red), treated with 200 nM rapamycin as indicated (orange bar). Scale bar, 2 μm. See *Figure 1—video 1*. (**C**) Representative confocal micrographs of HeLa cells before (light orange bar) and 2 min after 200 nM rapamycin treatment (dark orange bar). Cells expressing MitoTrap (red) with either our standard clathrin hook (FKBP-β2-GFP), clathrin-binding deficient mutant (FKBP-β2-Y815A/ΔCBM-GFP), or GFP-FKBP. In B and C, mitochondria were also labeled with MitoTracker Deep Red (blue) and orange arrowheads indicate MitoPits, where present. Inset, ×5 zoom. Scale bar, 10 μm. (**D**) Typical confocal micrograph of cells expressing FKBP-β2-GFP (green) and MitoTrap (red), treated with rapamycin (200 nM), fixed and stained with anti-PDHE2/E3 (blue). Inset, ×3 zoom. Scale bar, 5 μm. (**E**) Analysis of the spatial organization of MitoPits. (**i**) Line profile through the MitoPit shown in D, aligned to the FKBP-β2-GFP peak at 0 μm. Each of three channels is shown. (**ii**) Spatially averaged line profiles, aligned to the FKBP-β2-GFP peak at 0 μm, mean ± standard deviation (SD) is shown. (**iii**) Width of profiles for each channel in the dataset. (**iv**) Relative distance from the peak of FKBP-β2-GFP to the peak of MitoTrap (red) or PDHE2/3 (blue) for each profile in the dataset. Box plots indicate median, IQR, 9th and 91st percentiles. Each dot represents a profile.

The online version of this article includes the following video and figure supplement(s) for figure 1:

**Figure supplement 1.** Clathrin-coated pit formation on the ER, Golgi apparatus, and lysosomes.

**Figure 1—video 1.** The formation of MitoPits.

https://elifesciences.org/articles/78929/figures#fig1video1

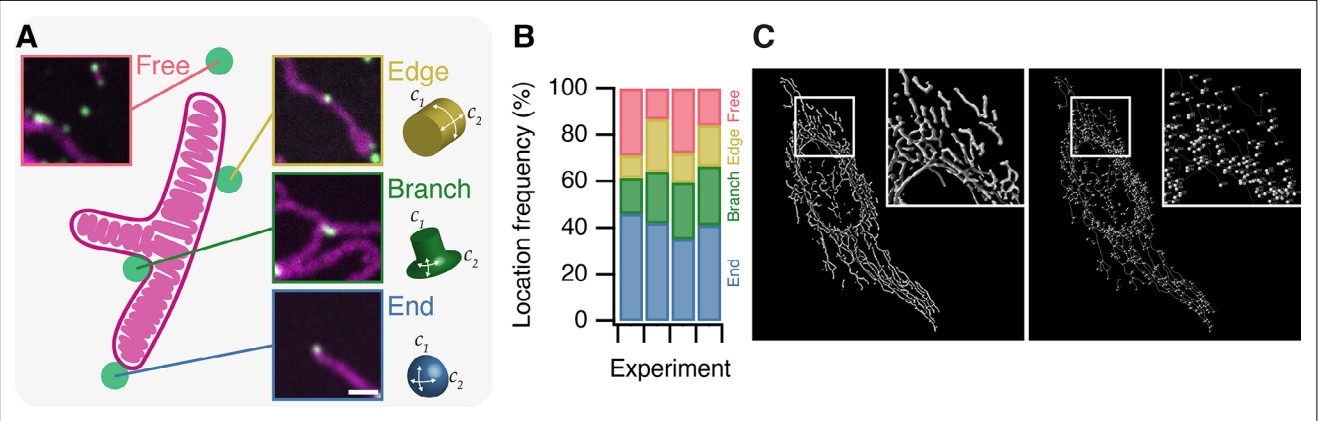

**Figure 2.** MitoPits form preferentially at mitochondria ends. (**A**) Schematic representation of a single-branched mitochondrion with representative micrographs of MitoPits that have formed at the indicated location in HeLa cells expressing FKBP-β2-GFP (green) and MitoTrap (red), treated with 200 nM rapamycin. Endpoints have positive curvature in two axes ($C_1$ and $C_2$), edges have only positive $C_1$ curvature, branchpoints have positive $C_1$ and negative $C_2$ curvature. Scale bar, 1 µm. (**B**) Frequency of MitoPits at each of the four locations across four independent experiments. $N_{spots}$ = 293, 223, 173, 297. (**C**) Typical MitoGraph from a HeLa cell expressing MitoTrap. 3D segmentation of the mitochondrial network (left) and network of edges and nodes (branchpoints and endpoints, right).

drawn through MitoPits perpendicular to the mitochondrial axis were taken on images where MitoPits had been induced and the cells stained for a mitochondrial matrix marker, PDHE2/3 (*Figure 1D, E*). Spatial averaging revealed a narrow distribution of FKBP-β2-GFP, consistent with the size of a CCP and a variable amount of mitochondrial anchor (*Figure 1Ei–iii*). The expected organization was evident with anchor and matrix preceding the hook by ~80 and ~180 nm, respectively (*Figure 1Eiv*).

Induction of presumptive CCPs was also achieved at the ER, Golgi, and lysosomes using compartment-specific membrane anchors fused to mCherry-FRB (*Figure 1—figure supplement 1*). Sec61β, Giantin TM domain (3131–3259), and LAMP1 were used for ER, Golgi, and lysosome anchors, respectively. Rerouting to the target membrane was seen for all hooks upon rapamycin addition, but spots only formed when using FKBP-β2-GFP as a clathrin hook and not with FKBP-β2 Y815A/ΔCBM-GFP nor GFP-FKBP (*Figure 1—figure supplement 1*). The formation of presumptive CCPs at all four intracellular locations suggests that our anchor-and-hook system is a transplantable module that can be used to induce the formation of CCPs at various locations. The mitochondrial outer membrane differs significantly in composition to other membranes in the cell and does not support any coated vesicle traffic; making this an ideal organelle to answer fundamental questions about CCV formation.

## MitoPits form preferentially at mitochondria ends

To begin characterizing MitoPits we first asked: do MitoPits form randomly over the mitochondrial surface? To address this question we recorded the location of MitoPits on 51 mitochondria from four independent experiments. A mitochondrial network can be described as a graph where edges are the cyclindrical surfaces, and nodes are either branchpoints or endpoints (*Figure 2*). We classified the frequency of MitoPits at these three locations in addition to Free MitoPits (not associated with a mitochondrion), which will be described later. Of the four locations, endpoints were the preferential site of MitoPit formation accounting for ~40% of the MitoPits (*Figure 2B*). When only considering endpoints and edges, we found that 72.7% of MitoPits were at endpoints. This indicates a preference for endpoints when compared with an expected 50/50 localization (odds ratio = 2.66, $\chi^2$ = 61.4, df = 1, p = 4.7 × 10$^{-15}$). However, this result is confounded by the fact that the mitochondrial network is dominated by edges. To correct for this, we computed mitochondrial graphs for six cells expressing MitoTrap and found that just 5.3% of the surface area is at endpoints versus 94.6% at edges (*Figure 2C*). Using this information, the corrected preference for endpoints over edges is 97.9% (odds ratio 47.3, $\chi^2$ = 541, df = 1, p < 2.2 × 10$^{-16}$). Endpoints have two axes of positive curvature, whereas edges only have one, which indicates a curvature preference for MitoPit formation. However, we note that branchpoints are a second-favored site of preferential formation (*Figure 2B*), and that a similar

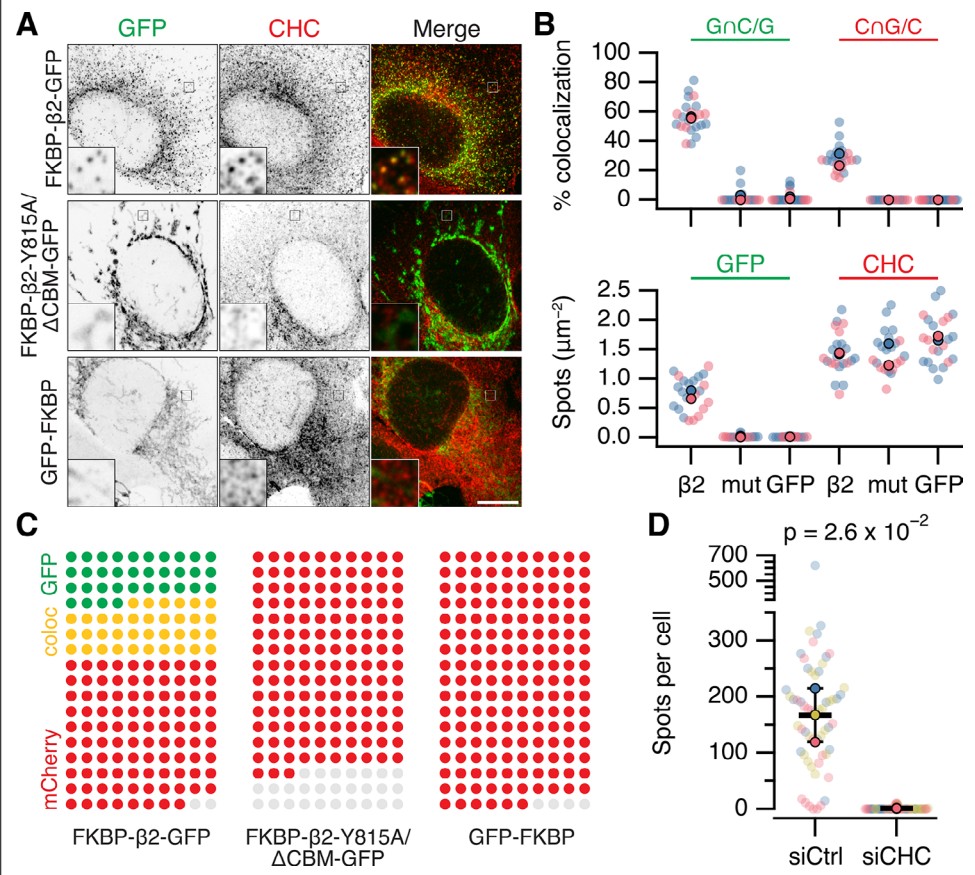

**Figure 3.** MitoPits are clathrin-coated pits. (**A**) Representative confocal micrographs of HeLa cells expressing dark MitoTrap with either clathrin hook (FKBP-β2-GFP), clathrin-binding deficient mutant hook (FKBP-β2-Y815A/ΔCBM-GFP), or GFP-FKBP. Cells were treated with 200 nM rapamycin before staining for clathrin heavy chain (CHC, red). Inset, ×5 zoom. Scale bar, 10 μm. (**B**) SuperPlots comparing colocalization (above) and spot density (below) for the conditions shown in A. Colors indicate the two independent experimental replicates. Each dot represents a cell, black outlined dots indicate the means of replicates. Spots of GFP or CHC were detected and quantified. Colocalization is shown as the percentage of GFP spots that coincided with CHC spots (left), or the percentage of CHC spots that coincided with GFP spots (right). Note, that this colocalization measure is likely an underestimate, but sufficient for comparison between conditions (see Materials and methods). (**C**) Waffle plots to visualize the median number of spots per 100 μm² that were positive for GFP only (green), clathrin only (red) or both (yellow), gray places indicate no spot. (**D**) SuperPlot to show the total number of FKBP-β2-GFP spots per cell after rapamycin addition in control (GL2, siCtrl) and CHC (siCHC) knockdown cells. Each dot represents a cell, black outlined dots indicate the means of replicates. p value is from Student's *t*-test with Welch's correction, *n* = 3.

The online version of this article includes the following figure supplement(s) for figure 3:

**Figure supplement 1.** MitoPits can be formed using diverse clathrin hooks.

argument applies here. These sites are saddle shaped with two axes of curvature, but of opposing polarity (*Figure 2A*). Our results indicate that MitoPits do not form at random locations but instead preferentially form at surfaces with specified geometry.

## MitoPits are clathrin coated and they can bud to form vesicles

To confirm that MitoPits are indeed CCPs on mitochondria, we used four different approaches. First, cells where MitoPits had been induced were stained for clathrin heavy chain (*Figure 3A*). Automated colocalization analysis revealed that ~60% of MitoPits were clathrin-positive while ~30% of clathrin-coated structures in the cell were MitoPits (*Figure 3B, C*). The density of MitoPits is about one-half of that of clathrin-coated structures in the cell (*Figure 3B, C*). Second, depletion of clathrin heavy chain using RNAi completely inhibited the formation of MitoPits (*Figure 3D*). Third, MitoPits could

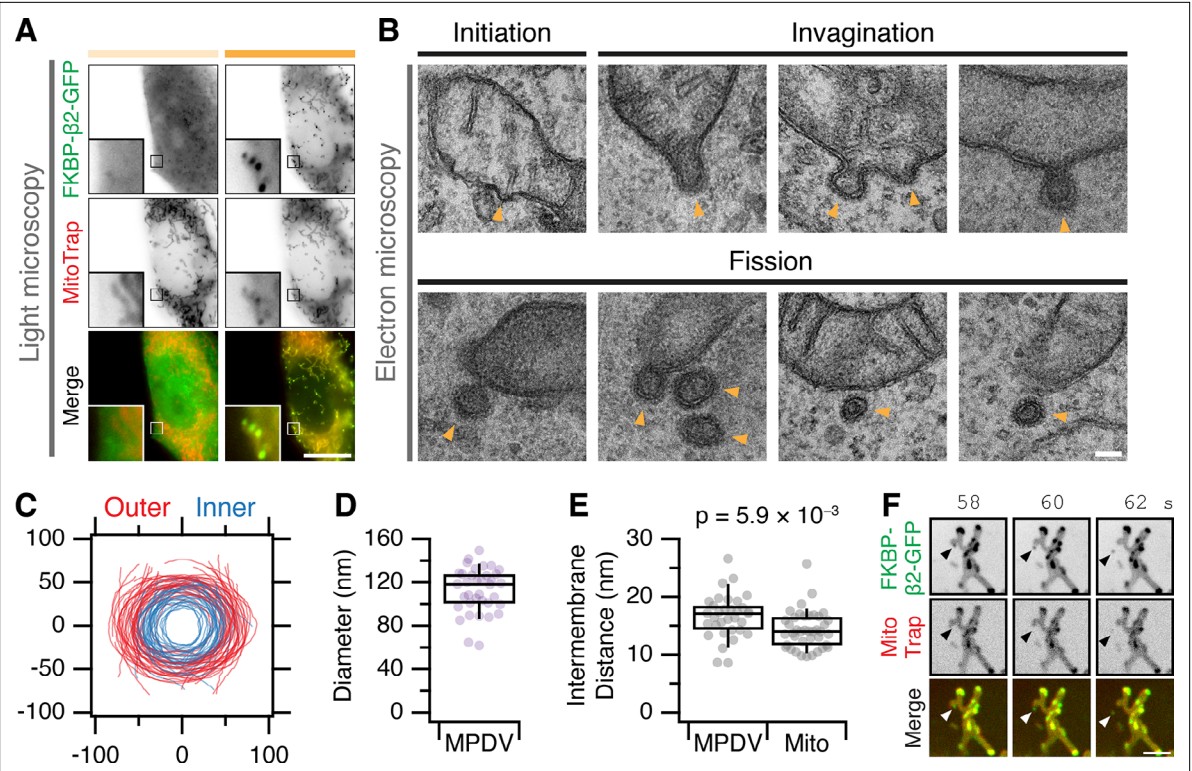

**Figure 4.** MitoPits are clathrin coated and can bud to form vesicles. (**A**) Fluorescence micrographs of a HeLa cell expressing FKBP-β2-GFP (green) and MitoTrap (magenta), before (left) and after (right) treatment with 200 nM rapamycin. Insets, ×5 zoom. Scale bar, 10 μm. (**B**) Electron micrographs of ultrathin (80 nm) sections taken from the cell shown in A. All morphological stages of clathrin-coated pit formation were observed (indicated by arrowheads) including evidence of fission of MitoPit-derived vesicles (MPDVs). Scale bar, 100 nm. (**C**) Membrane profiles of 36 MPDVs rotated so that the major axis of the outer membrane is at y = 0. (**D, E**) Box plots to show the average outer diameter and the intermembrane distance of MPDVs. Each dot represents a MPDV, taken from three cells. Box plots indicate median, IQR, 9th and 91st percentiles. p value from Student's *t*-test. (**F**) Stills from a live cell imaging experiment (see **Figure 4—video 1**). MitoPit formation was induced by 200 nM rapamycin addition after 8 s. Arrowheads show the formation of a MitoPit that buds to form a distinct vesicle. Scale bar, 1 μm.

The online version of this article includes the following video for figure 4:

**Figure 4—video 1.** Budding of MitoPit-derived vesicles.

https://elifesciences.org/articles/78929/figures#fig4video1

be formed using other clathrin-binding domains as clathrin hooks. FKBP-β1-GFP, FKBP-AP180-GFP, or FKBP-epsin-GFP were competent, like FKBP-β2-GFP, for MitoPit formation (**Figure 3—figure supplement 1B**). In contrast, FKBP-α-GFP or FKBP-β3-GFP were inactive as GFP-FKBP (**Figure 3—figure supplement 1A**). These results, together with the observation that clathrin-binding deficient mutant FKBP-β2-GFP (Y815A/ΔCBM) does not form MitoPits indicate that clathrin binding is essential for MitoPit formation (**Figures 1C and 3A**).

Our fourth approach was to directly observe MitoPits by electron microscopy (**Figure 4**). We imaged cells by light microscopy in which MitoPits were induced, and then located the same cell for processing for EM (**Figure 4A**). MitoPits were readily observable in electron micrographs of ultrathin (80 nm) sections (**Figure 4B**). All morphologically defined stages of CCP formation were seen on the mitochondria from initiation through to shallow and deep invagination, including neck formation. Notably, both the inner and outer mitochondrial membranes were deformed together in the MitoPit, indicating the tight linkage between these two membranes. Although these deformations are technically *evaginations* of the mitochondria, we refer to them as *invaginations* (toward the cytoplasm) for consistency. MitoPits had an unmistakable electron-dense coat typical of a CCP (**Figure 4B**). To our surprise, we also imaged many examples indicative of fission: clathrin-coated double-membrane vesicles in close proximity to, but distinct from, a mitochondrion. These MitoPit-derived vesicles (MPDVs) were ~120 nm diameter on average (**Figure 4C, D**), and the space between inner and outer membranes

in the MPDV was slightly larger than the mitochondrial intermembrane distance (*Figure 4E*). Although the deeply invaginated pits suggested fission, it is possible that in a 80 nm section, the vesicle may still be attached to a mitochondrion which is out-of-section. We returned to live cell imaging of MitoPit formation and were able to observe MPDVs budding from the mitochondrial surface (*Figure 4F*). In some cases, multiple budding events could be visualized from the same endpoint location on the mitochondrion (*Figure 4—video 1*). These experiments confirm that MitoPits are CCPs and indicate that MitoPits can bud to form CCVs.

## Pinchase-independent formation of MPDVs

We next sought to determine the scission factor (pinchase) responsible for MPDV fission. The leading candidate was dynamin, given its role in scission during endocytosis at the plasma membrane (*Antonny et al., 2016*). Three distinct approaches were used to test the involvement of dynamin in MPDV fission. In all cases our assay was simply to quantify the fraction of induced FKBP-β2-GFP spots that were free MPDVs (*Figure 2*, see Methods).

First, overexpression of dominant-negative mutant dynamin-1(K44A)-mCherry (*Van der et al., 1993*) was used in cells expressing FKBP-β2-GFP and dark MitoTrap, and compared with no expression or overexpression of dynamin-1-mCherry (*Figure 5A, E*). In all conditions, a similar percentage of MPDVs as a fraction of total FKBP-β2-GFP spots was measured in all three conditions.

Second, cells expressing FKBP-β2-GFP and dark MitoTrap were treated with dynamin inhibitor Dynole 34–2 (30 µM) or a control compound (Dynole 31–2) prior to and during MitoPit formation. A similar fraction of MPDVs were observed in Dynole 34–2-treated cells when compared with the control (*Figure 5B, F*). We confirmed in the same cells that Dynole 34–2 treatment had impaired endocytosis of transferrin, showing that dynamin activity was successfully inhibited in the cells where MPDVs were formed (*Figure 5B*). This result made us question whether something about our minimal system for inducing CCP formation precluded dynamin participation. However, when using FKBP-β2-GFP and CD8-dCherry-FRB to induce endocytosis at the plasma membrane, we saw inhibition using Dynole 34–2 (*Figure 5—figure supplement 1*). This result suggested that if dynamin were involved in MPDV fission, we should have an effect with chemical inhibition. Nonetheless it was formally possible that Dynole and dominant-negative approaches had not inhibited the specific isoform of dynamin that causes fission of MPDVs.

Third, fibroblasts from a dynamin triple knockout (TKO) mouse were used to test if any of the three dynamins were involved in MPDV formation (*Ferguson et al., 2009*; *Park et al., 2013*). Conditional dynamin TKO was induced prior to expression of FKBP-β2-GFP and dark MitoTrap and induction of MitoPit formation (*Figure 5C, G*). Again, there was no significant difference in the percentage of MPDVs formed between the control and TKO cells (*Figure 5C*). Visualization of transferrin uptake in the same cells confirmed that endogenous endocytosis was inhibited in dynamin TKO cells.

Together our results, using three independent approaches to inhibit dynamin activity, indicate that dynamin is not responsible for fission of MitoPits into MPDVs.

We next considered alternative pinchase candidates, starting with Drp1 (Dynamin-1-like protein, DNM1L) the mitochondria-specific fission enzyme (*Ingerman et al., 2005*). MitoPit formation was induced in cells expressing mCherry-Drp1 or a dominant-negative version (mCherry-Drp1 K38A) together with FKBP-β2-GFP and dark MitoTrap. Overexpression of WT or mutant Drp1 resulted in fragmented or hyperfused mitochondria, respectively, demonstrating that the Drp1 constructs worked as expected (*Figure 5D*). If Drp1 is responsible for MPDV budding, we would expect to see a reduction in free spots in cells expressing mCherry-Drp1 K38A; however, the percentage was similar to nonexpressing cells (*Figure 5H*). In fact, we measured a small *increase* in the percentage of MPDVs in cells expressing mCherry-Drp1 WT. This result can be explained by the fragmentation of mitochondria caused by overexpression of WT Drp1 giving rise to more mitochondrial ends, and our earlier observation that MitoPits form preferentially at endpoints (*Figure 2*). However, the lack of inhibition of MPDV formation with the dominant-negative Drp1 mutant suggests that Drp1 is not involved in the fission of MitoPits.

We also investigated a role for ESCRT-III or actin in MPDV budding, given their role in other membrane scission events, yet found no evidence that either were involved in fission (*Figure 5—figure supplement 2*). Our conclusion is that none of the usual pinchase candidates are responsible

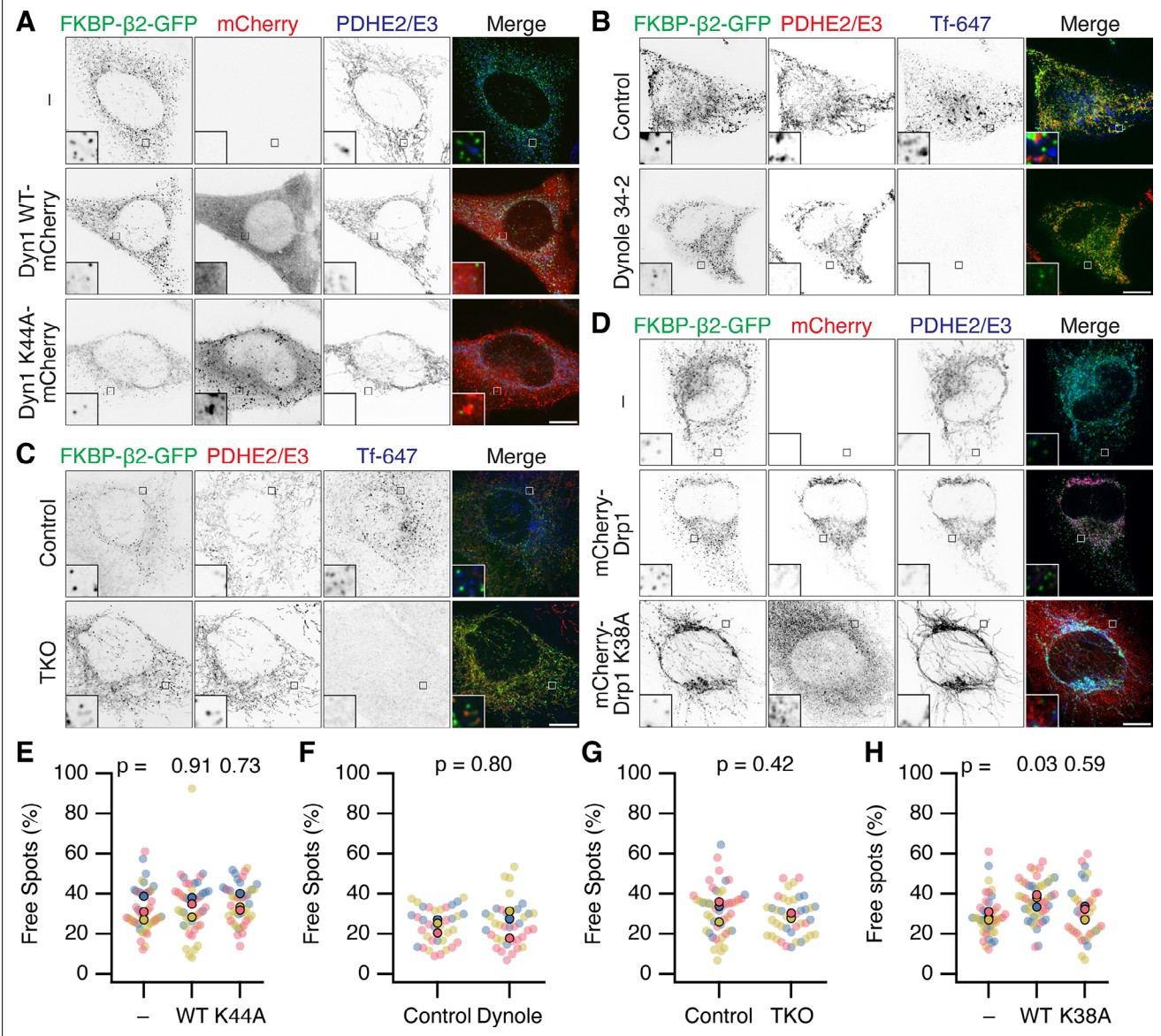

**Figure 5.** Neither Dynamin nor Drp1 activity is required for scission of MitoPit-derived vesicles. (**A–D**) Representative confocal micrographs of MitoPit-derived vesicle formation under different approaches to inhibit dynamin or Drp1 function. (**A**) Dominant-negative dynamin-1: HeLa cells expressing dark MitoTrap and FKBP-β2-GFP (green) alone (–) or in combination with either Dyn1 WT-mCherry WT or Dyn1 K44A-mCherry (red), treated with rapamycin (200 nM, 30 min), stained with anti-PDHE2/E3 (blue). (**B**) Chemical inhibition: HeLa cells expressing dark MitoTrap and FKBP-β2-GFP (green), treated with control compound (Dynole 31–2, 30 μM) or Dynole 34–2 (30 μM) for 25 min and rapamycin (200 nM) for the final 10 min. Fluorescent human transferrin (Tfn 647, blue) indicates endocytic activity, mitochondrial matrix was stained with anti-PDHE2/E3 (red). (**C**) Dynamin triple knockout: inducible dynamin TKO mouse embryonic fibroblasts expressing dark MitoTrap and FKBP-β2-GFP (green), treated with vehicle (Control) or 3 μM Tamoxifen (TKO) for 2 days prior to transfection and with rapamycin (200 nM, 10 min) for clathrin-coated pit (CCP) induction. Fluorescent human transferrin (Tfn 647, blue) indicates endocytic activity, mitochondrial matrix was stained with anti-PDHE2/E3 (red). (**D**) Dominant-negative Drp1: HeLa cells expressing dark MitoTrap and FKBP-β2-GFP (green) alone (–) or in combination with either mCherry-Drp1 WT or mCherry-Drp1 K38A (red), treated with rapamycin (200 nM, 30 min), stained with anti-PDHE2/E3 (blue). Insets, ×5 zoom. Scale bars, 10 μm. (**E–H**) SuperPlots showing the percentage of free spots for each condition. Colors represent replicates, dots represent cells, solid dots represent the mean of each replicate. Indicated p values from Dunnett's post hoc test (**E, H**) or Student's *t*-test (**F, G**).

The online version of this article includes the following figure supplement(s) for figure 5:

**Figure supplement 1.** Hot-wired endocytosis requires dynamin activity.

**Figure supplement 2.** No evidence for ESCRTIII or actin involvement in fission of MitoPits.

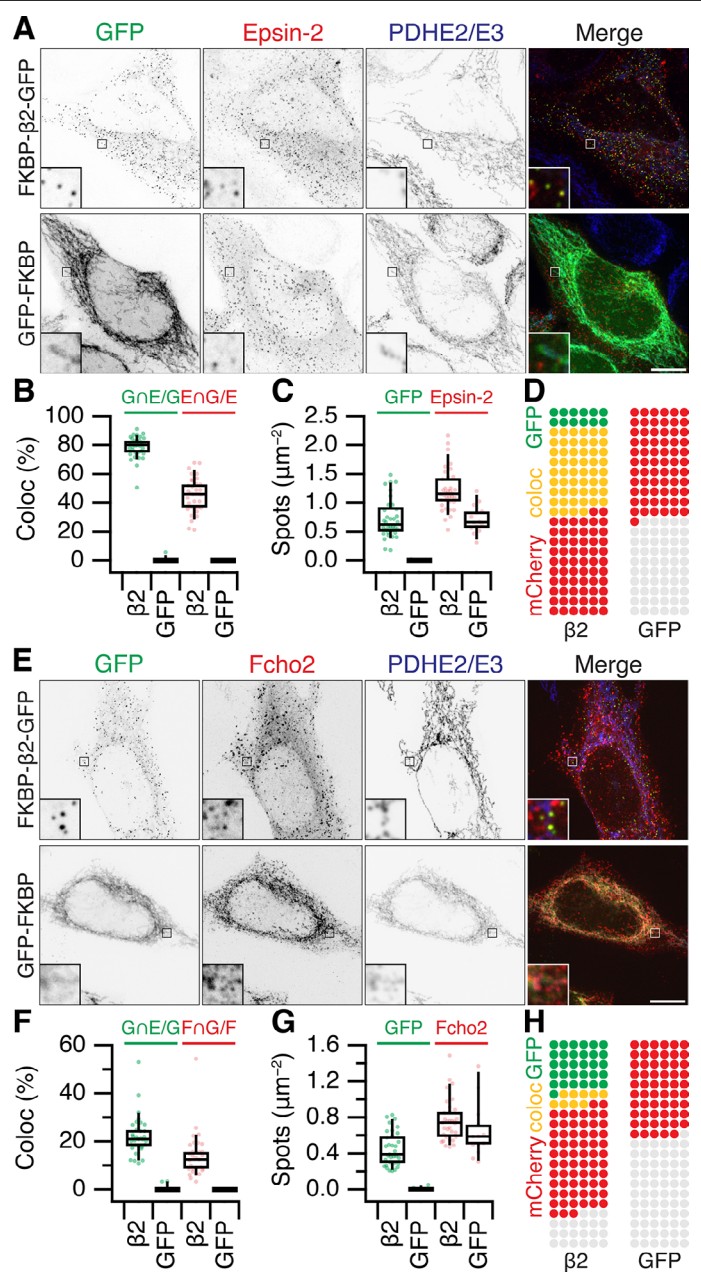

**Figure 6.** Accessory protein recruitment to MitoPits. Representative confocal micrographs of HeLa cells expressing dark MitoTrap and mCherry- Epsin-2 (**A**) or mCherry-Fcho2 (**E**) (red) with either clathrin hook (FKBP-β2-GFP) or GFP-FKBP (green). Cells were treated with 200 nM rapamycin before staining for PDHE2/E3 (blue). Inset, ×5 zoom. Scale bars, 10 μm. Box plots to compare colocalization (**B, F**) and spot density (**C, G**). Each dot represents a cell. Box plots indicate median, IQR, 9th and 91st percentiles. Spots of GFP and mCherry were detected and quantified. Colocalization is shown as the percentage of GFP spots that coincided with Epsin-2 or Fcho2 spots (left), or the percentage of Epsin-2 or Fcho2 spots that coincided with GFP spots (right). (**D, H**) Waffle plots to visualize the median number of spots per 100 μm² that were positive for GFP only (green), clathrin only (red), or both (yellow).

The online version of this article includes the following figure supplement(s) for figure 6:

**Figure supplement 1.** MitoPits do not recruit AP1, AP2, AP3, amphiphysin-1, endophilin-1, Hip1R, or SNX9.

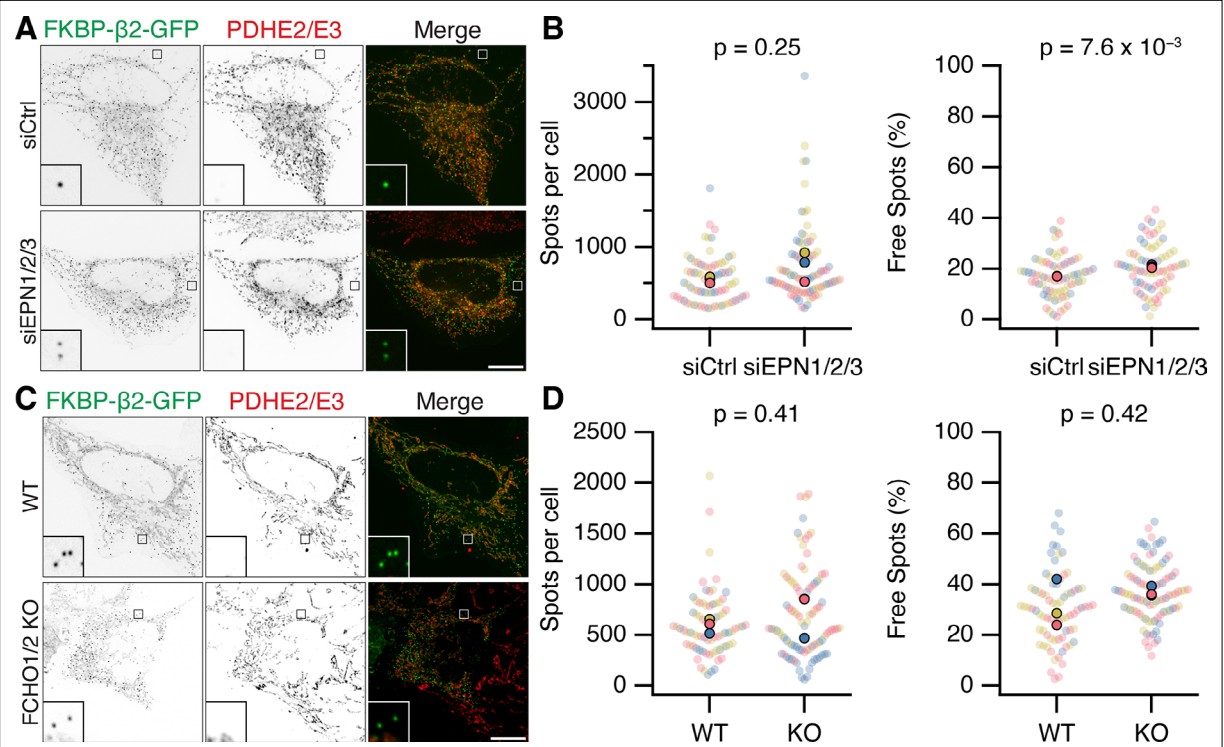

**Figure 7.** Epsins and FCHO proteins are dispensable for MitoPit formation and vesicle generation. (**A, C**) Representative confocal micrographs of HeLa cells expressing dark MitoTrap and clathrin hook (FKBP-β2-GFP, green). In A, cells were transfected with control (siCtrl) or triple Epsin (siEPN1/2/3) siRNAs; in C, either wild-type or FCHO1/2 knockout (KO) HeLa cells were used. Cells were treated with 200 nM rapamycin before staining for PDHE2/E2 (red). Inset, ×5 zoom. Scale bars, 10 μm. (**B, D**) SuperPlots showing the spots per cell or the percentage of free spots for each condition. Colors represent replicates, dots represent cells, solid dots represent the mean of each replicate. Indicated p values from Student's *t*-test.

for MPDV budding which suggests that MPDV release most likely occurs via a pinchase-independent mechanism, most likely a passive fission process (*Renard et al., 2018*).

## A minimal machinery for intracellular CCV formation

Are MitoPits exclusively composed of anchor, hook, and clathrin? To address this question we carried out an imaging survey for accessory proteins that might be recruited to MitoPits (*Figure 6* and *Figure 6—figure supplement 1*). Accessory proteins tagged with red fluorescent proteins were expressed in cells along with dark MitoTrap and FKBP-β2-GFP or GFP-FKBP. MitoPit formation was triggered by rapamycin addition and the recruitment of accessory proteins to green spots was assessed. We found no recruitment of AP-1 (σ1-mCherry), AP-2 σ2-mCherry, AP-3 (σ3-mCherry), amphiphysin (mCherry-amphiphysin), Endophilin-A1 (Endophilin-RFP), Huntingtin-interacting protein 1-related protein (Hip1R-tDimer-RFP), or sorting nexin-9 (mCherry-SNX9) (*Figure 6—figure supplement 1*). In all cases, the distribution of the accessory protein was the same in cells with and without MitoPits and no colocalization was seen between the accessory and MitoPits. However, two accessory proteins, Epsin-2 (mCherry-Epsin-2) and F-BAR domain only protein 2 (mCherry-Fcho2) *were* recruited to MitoPits (*Figure 6*). Both proteins have been implicated in membrane bending during the early stages of endocytosis. In the case of Epsin-2, ~80% of MitoPits were Epsin-2-positive, whereas only ~20% of MitoPits had Fcho2 (*Figure 6B, F*).

It is perhaps not surprising that concentrating the hook and clathrin at specific locations causes the recruitment of some proteins that can bind either one. However, it was important to address whether Epsins and FCHO proteins were required for MitoPit formation. Using RNAi depletion of three Epsins (Epsin-1, Epsin-2, and Epsin-3), we found that the number of MitoPits per cell was equivalent to control RNAi (*Figure 7A, B*). Similarly, HeLa cells where both FCHO1 and FCHO2 were knocked out had similar numbers of MitoPits to control HeLa cells (*Figure 7C, D*). Moreover, analysis of the number of free spots showed that there was no decrease in the number of MPDVs formed following

depletion of Epsins or knockout of FCHO1/2 (*Figure 7B, D*). These results indicate that Epsins and FCHO proteins are bystanders: recruited to the site of MitoPits but are not required functionally for MitoPit formation or fission.

## Discussion

In this study, we showed that CCPs can be induced to form on intracellular membranes using minimal components. The induction of MitoPits was of particular interest since mitochondria do not normally support clathrin-mediated traffic and they have a different composition to the plasma membrane, with 10-fold less PI(4,5)P$_2$ which is an important phophoinositide for the function of many endocytic accessory proteins. All stages of CCV formation were recapitulated, including fission, which occurred without the activity of known pinchases. Our findings suggest that, in this context, clathrin may act in the absence of accessory proteins to generate CCVs.

MPDVs were classified as MitoPit spots that were free from mitochondria (~20% of the total). This fraction was unchanged when the activity of four scission candidates was compromised. Interference with dynamin, Drp1, Vps4a, or the actin cytoskeleton, using a variety of approaches, had no effect on the proportion of free spots suggesting that fission of MPDVs does not require a scission molecule. A possible confounder to this result is that our analysis methods may detect free spots that are not MPDVs. First, small mitochondrial fragments may house a MitoPit giving the impression of a free spot. However, fragmented mitochondria are much larger than MitoPits, and are excluded by the upper limit of our detection method. Second, endogenous mitochondrially derived vesicles (MDVs) may conflate the analysis. MDVs are very rare, with only 5–7 MDVs per cell on average (*Neuspiel et al., 2008*), whereas our synthetic system creates hundreds of MitoPits per cell; therefore any contribution of MDVs to the quantification of MPDVs will be negligible. In any case, MitoPits form synchronously after induction and free spots can be seen to bud from the mitochondria later, which means the possibility of such misclassification is very low.

The apparent lack of scission factor for MPDVs is surprising given the canonical role of dynamin in CME and the involvement of Drp1 in MDV formation (*König et al., 2021*). On the other hand, GTP-hydrolyzing scission molecules may not be essential for intracellular budding events generally. For example, the fission of COPI- and COPII-coated vesicles does not require a specific scission molecule (*Adolf et al., 2013*) and imaging studies indicate that intracellular AP1/clathrin budding events might occur without dynamin activity (*Kural et al., 2012*). There are many examples of vesicle budding events that do not require an active- or scission-based mechanism for fission to occur (*Renard et al., 2018*). Our model therefore is that the clathrin coat is sufficient to deform the membrane and also to cause its fission. If intracellular CCV formation does not require a pinchase, and this requirement is exclusive to the plasma membrane, an interesting question for the future is: what is special about the plasma membrane that means that a pinchase is required?

Out of three AP complexes and six endocytic accessory proteins, only Epsin-2 and FCHo2 were recruited to MitoPits. Epsin-2 recruitment was stronger, appearing at ~80% of MitoPits versus ~20% for Fcho2. This is explained by Epsin-2 potentially binding to the β2 appendage of the hook as well as clathrin (*Owen et al., 2000*; *Drake et al., 2000*), whereas Fcho2 is not known to bind directly to either, and is potentially recruited via Eps15 (*Reider et al., 2009*). We showed that Epsins and FCHO proteins are apparently bystanders: they were recruited to these sites of concentrated clathrin, but did not functionally participate in MitoPit formation. Their lack of function may be due to (1) being recruited later than their normal temporal window of function, (2) failure to engage with the mitochondrial membrane, or (3) lack of components to interact with in our minimalist system. The implication is that clathrin is the primary factor that deforms the mitochondrial membrane.

Clathrin, working as a Brownian ratchet, has been proposed to drive membrane bending during CCP formation (*Hinrichsen et al., 2006*; *Dannhauser and Ungewickell, 2012*), although the contribution of membrane bending proteins in the process has been difficult to dissect (*Sochacki and Taraska, 2019*). Multiple lines of evidence demonstrate that MitoPit formation is clathrin dependent: (1) colocalization of clathrin with MitoPits, (2) direct visualization of a clathrin coat by EM, (3) the absence of MitoPit induction when clathrin heavy chain was downregulated by RNAi, and (4) MitoPit formation could be initiated with a variety of hooks that can effectively bind clathrin such as β1, Epsin, and AP180-C. The clathrin hooks that we use all contain intrinsically disordered regions and such proteins have been shown to deform membranes via a phase separation mechanism (*Busch et al., 2015*; *Yuan*

*et al., 2021*). Importantly, a clathrin-binding deficient β2 hook which differs from the wild-type by only a few residues was unable to support MitoPit formation, arguing against a contribution from the hook alone via this mechanism. It is tempting therefore to conclude that curvature generation in the context of MitoPits is by clathrin alone. However, it is possible that the disordered region of the hook contributes to curvature generation in a manner that is organized by clathrin, i.e. that the clathrin lattice could spatially constrain the hook and, when concentrated, the disordered regions contribute to curvature. Separating these possibilities is difficult since both clathrin and hook are essential to make MitoPits. Whatever the mechanism, it seems that enough force is generated by our synthetic system to deform both the inner and outer mitochondrial membranes and even to pinch off the MitoPits.

Out of three AP complexes and six endocytic accessory proteins, only Epsin-2 and FCHo2 were recruited to MitoPits. Epsin-2 recruitment was stronger, appearing at ~80% of MitoPits versus ~20% for Fcho2. This is explained by Epsin-2 potentially binding to the β2 appendage of the hook as well as clathrin (*Owen et al., 2000*; *Drake et al., 2000*), whereas Fcho2 is not known to bind directly to either, and is potentially recruited via Eps15 (*Reider et al., 2009*). We showed that Epsins and FCHO proteins are apparently bystanders: they were recruited to these sites of concentrated clathrin, but did not functionally participate in MitoPit formation. Their lack of function may be due to (1) being recruited later than their normal temporal window of function, (2) failure to engage with the mitochondrial membrane, or (3) lack of components to interact with in our minimalist system. The implication is that clathrin is the primary factor that deforms the mitochondrial membrane.

A rough calculation indicates that this mechanism is energetically feasible, even for a double-membraned vesicle. The energy per unit of membrane area required to form a spherical vesicle can be thought of as the sum of the bending energy ($G_{bending}$), membrane tension ($\gamma$), and cargo crowding (*Stachowiak et al., 2013*). $G_{bending}$ is described in *Equation 1*,

$$G_{bending} = \frac{8\pi\kappa}{4\pi r^2} = \frac{2\kappa}{r^2} \tag{1}$$

where $\kappa$ is the bending rigidity, $r$ is the vesicle radius, and $G_{bending}$ is in units of $k_BT$ nm$^{-2}$ (where $k_BT$ is the thermal energy and is ~4 × 10$^{-21}$ J mol$^{-1}$ or ~4.3 pN nm). Typical values for $G_{bending}$ are 10–50 $k_BT$ (*Bochicchio and Monticelli, 2016*) while $\gamma$ at the plasma membrane is 0.02 $k_BT$ nm$^{-2}$ (*Evans and Rawicz, 1990*). The mitochondrial membrane is more fluid than the plasma membrane due to its low sterol and high cardiolipin content (*Horvath and Daum, 2013*), and it does not have the same osmotic imbalance and actin interactions which elevate $\gamma$. Indeed, physical measurements of membrane tension of the outer mitochondrial membrane indicate $\kappa$ = 15 $k_BT$, and membrane tension to be 0.0025 $k_BT$ nm$^{-2}$ (*Gonzalez-Rodriguez et al., 2015*; *Wang et al., 2008*). Therefore, the cost of making a double-membraned vesicle with a 60 nm radius as observed in our EM images, can be estimated as $2 \times \left(2\kappa/r^2 + \gamma\right) = 2 \times \left(30/r^2 + 0.0025\right) \approx 0.022$. Assembled clathrin can contribute an estimated 0.08 $k_BT$ nm$^{-2}$ (*Stachowiak et al., 2013*; *den Otter and Briels, 2011*). Although the cost of cargo crowding has not been included, this suggests that clathrin assembly by itself would be sufficient to account for MPDV formation. However, cargo crowding may be substantial, and interactions between inner and outer membranes may also increase the cost significantly. This would invoke the need for clathrin-organized crowding of intrinsically disordered hooks to assist with curvature generation. Comparison between this calculation and the energetics of CCP formation at the plasma membrane is difficult due to a number of other factors (see *Hassinger et al., 2017* for a detailed analysis).

This calculation assumes the formation of a spherical vesicle from a flat sheet. We observed that MitoPits form preferentially at branchpoints or endpoints on the mitochondrial surface, likely indicating a geometry preference for clathrin assembly on dual curvature surfaces (*Larsen et al., 2020*). Even the MitoPits observed on mitochondrial edges are formed at a site of single curvature. The size of a CCP ($r$ = 60 nm) is ~4 times smaller than a mitochondrial endpoint ($r$ = 250 nm). So whilst considerable deformation is still required to form a vesicle at these precurved sites, the total energy requirement is lower than at a flat sheet. This preference for precurved sites echoes classic work showing CCPs forming at the curved edges of intracellular membranes (*Aggeler et al., 1983*), while a very recent study suggests that clathrin prefers precurved surfaces (*Zeno et al., 2021*).

Interestingly, cells grown on nanofabricated surfaces that induce inward curvature of the plasma membrane in ridges show increased CCP formation at these pre-curved sites (*Zhao et al., 2017*; *Cail et al., 2022*). The ridges have widths of 75–500 nm, and significant enrichment of adaptors and clathrin is seen at 200 nm and narrower (*Cail et al., 2022*). The width of a mitochondrion is ~500 nm,

so the MitoPits that form on the edges of mitochondria in our study are not in the range where enrichment has been observed due to induced curvature at the plasma membrane. In addition, the pits formed in that study scale with the size of the precurved site whereas MitoPits and MPDVs are of a uniform size that has a tighter curvature than that of the mitochondrion surface. So while more extreme curvatures promote spontaneous (and clathrin-independent) vesicle formation at the plasma membrane (*Cail et al., 2022*) the situation for MitoPits is different, with formation being triggered by the experimenter and the curvature of the underlying mitochondrial membrane likely assisting MPDV formation, but not prespecifying their shape.

Which aspects of our work can translate to CCP formation at the plasma membrane during CME? Besides a scission requirement, there are three other differences between MitoPits and CME. First, the CCVs emerge from a flattened lattice that has the propensity to remodel (*Heuser, 1989*; *Lin et al., 1991*; *Sochacki et al., 2021*). It is unclear if flat lattices are formed at the mitochondria after clathrin is recruited; the mitochondrial outer membrane presumably lacks many key features of the ventral cell surface. However, the spontaneous remodeling of flat lattices occurs at the plasma membrane without added accessory factors (*Sochacki et al., 2021*), which is consistent with our clathrin-centric model. Second, actin is actively involved in force generation in some circumstances at the plasma membrane (*Hassinger et al., 2017*; *Kaur et al., 2014*; *Taylor et al., 2011*), whereas we found no requirement for MPDV formation. Third, CCPs at the plasma membrane are spatially constrained (*Aguet et al., 2013*), whereas MitoPits, particularly on mitochondrial edges, are motile. Despite these differences, CME at the plasma membrane can be triggered in an analogous way (*Wood et al., 2017*) and previous in vitro work supports the idea of a minimal machinery for CCV formation (*Dannhauser and Ungewickell, 2012*).

Our work suggests that from the moment of recruitment onwards, clathrin is sufficient to form a vesicle. What does this mean for the network of accessory proteins associated with the core clathrin machinery? The clathrin-centric mechanism we describe suggests that none of these proteins are 'mediators' of vesicle formation, and instead they may act as 'modulators'; enhancing vesicle formation, changing vesicle size or adapting it to certain conditions. It is important to note that in our system, we trigger the recruitment of clathrin to a precurved intracellular surface using a hook that may be active, and we are blind to potential mediators (cargo, lipids, and other proteins) acting earlier. However, since clathrin recruitment defines the first stage of CCV formation – initiation – the proposed mechanism accounts for almost the entire pathway. Given the differences between intracellular CCV formation and CME, the distinction between mediators and modulators at the plasma membrane may differ slightly.

Our system, of ectopic placement of a clathrin-binding domain and the subsequent action of endogenous clathrin, represents a transplantable module for CCP formation at potentially any membrane. As well as allowing fundamental questions about vesicle formation in cells to be addressed, this system may be used in the future to manipulate the size and composition of target organelles and dissect intracellular processes that are experimentally inaccessible.

## Materials and methods

### Key resources table

| Reagent type (species) or resource | Designation | Source or reference | Identifiers | Additional information |
|---|---|---|---|---|
| Antibody | Anti-clathrin heavy chain (X22). Mouse monoclonal | ATCC | RRID:CVCL_F814 | 1:1000 |
| Antibody | Pyruvate dehydrogenase E2/E3bp antibody. Mouse monoclonal | Abcam | RRID:AB_10862029 | 1 µg/ml |
| Cell line (*H. sapiens*) | FCHO1/2 KO HeLa | *Umasankar et al., 2014* | #64/1.E | |
| Cell line (*M. musculus*) | Dynamin triple knockout cells | *Park et al., 2013* | DNM TKO | |
| Chemical compound, drug | Acti-stain 555 phalloidin | Cytoskeleton, Inc. | #PHDH1-A | Final 1:1000 |
| Chemical compound, drug | Dynamin Inhibitors: Dynole Series Kit | Abcam | ab120474 | 30 µM |
| Chemical compound, drug | Latrunculin B | Sigma-Aldrich | #428,020 | 1 µM |

*Continued on next page*

*Continued*

| Reagent type (species) or resource | Designation | Source or reference | Identifiers | Additional information |
|---|---|---|---|---|
| Chemical compound, drug | Rapamycin | Alfa Aesar | J62473 | 200 nM |
| Other | MitoTracker Deep Red FM | Thermo Fisher | M22426 | 1:15,000 |
| Other | Transferrin From Human Serum, Alexa Fluor 647 Conjugate | Thermo Fisher | T23366 | 100 µg/ml |
| Recombinant DNA reagent | CD8-dCherry-FRB | *Wood et al., 2017* | RRID:Addgene_100739 | |
| Recombinant DNA reagent | Dyn1 K44A-mCherry | This paper | - | See Materials and methods - Molecular Biology section. Modification of Addgene plasmid #34681. GFP replaced by mCherry. Available from the corresponding author. |
| Recombinant DNA reagent | Dyn1 WT-mCherry | This paper | - | See Materials and methods - Molecular Biology section. Modification of Addgene plasmid #34680. GFP replaced by mCherry. Available from the corresponding author. |
| Recombinant DNA reagent | Endophilin-1–247-RFP | L.Lagnado, University of Sussex | - | |
| Recombinant DNA reagent | FKBP-AP180-GFP | This paper | RRID:Addgene_186577 | see Materials and methods - Molecular Biology section |
| Recombinant DNA reagent | FKBP-epsin-GFP | *Wood et al., 2017* | RRID:Addgene_100733 | |
| Recombinant DNA reagent | FKBP-α-GFP | *Wood et al., 2017* | RRID:Addgene_100731 | |
| Recombinant DNA reagent | FKBP-β1-GFP | *Wood et al., 2017* | RRID:Addgene_100732 | |
| Recombinant DNA reagent | FKBP-β2 Y815A/ΔCBM-GFP | *Wood et al., 2017* | RRID:Addgene_100729 | |
| Recombinant DNA reagent | FKBP-β2-GFP | *Wood et al., 2017* | RRID:Addgene_100726 | |
| Recombinant DNA reagent | FKBP-β3-GFP | *Wood et al., 2017* | RRID:Addgene_100734 | |
| Recombinant DNA reagent | FRB-mCherry-Giantin | This paper | RRID:Addgene_186575 | see Materials and methods - Molecular Biology section |
| Recombinant DNA reagent | FRB-mCherry-Sec61β | This paper | RRID:Addgene_186574 | see Materials and methods - Molecular Biology section |
| Recombinant DNA reagent | GFP-FKBP | *Wood et al., 2017* | - | |
| Recombinant DNA reagent | Hip1R-tDimer-RFP | Addgene | RRID:Addgene_27700 | |
| Recombinant DNA reagent | Lamp1-mCherry-FRB | This paper | RRID:Addgene_186576 | see Materials and methods - Molecular Biology section |
| Recombinant DNA reagent | mCherry-amphiphysin 1 | Addgene | RRID:Addgene_27692 | |
| Recombinant DNA reagent | mCherry-Drp1 | Addgene | RRID:Addgene_49152 | |
| Recombinant DNA reagent | mCherry-Drp1 K38A | This paper | - | See Materials and methods - Molecular Biology section. Modification of Addgene plasmid. K38A mutation introduced into mCherry-Drp1. Available from the corresponding author. |
| Recombinant DNA reagent | mCherry-epsin2 | Addgene | RRID:Addgene_27673 | |
| Recombinant DNA reagent | mCherry-FCHo2 | Addgene | RRID:Addgene_27686 | |
| Recombinant DNA reagent | mCherry-SNX9 | Addgene | RRID:Addgene_27678 | |
| Recombinant DNA reagent | mCherry-Vps4a E228Q | This paper | - | See Materials and methods - Molecular Biology section. mCherry-tagged Vps4a E228Q (converted from GFP-Vps4a E228Q). Available from the corresponding author. |
| Recombinant DNA reagent | mCherry-Vps4a WT | This paper | - | See Materials and methods - Molecular Biology section. mCherry-tagged Vps4a (converted from GFP-Vps4a). Available from the corresponding author. |

*Continued on next page*

*Continued*

| Reagent type (species) or resource | Designation | Source or reference | Identifiers | Additional information |
|---|---|---|---|---|
| Recombinant DNA reagent | pMito-dCherry-FRB | This paper | RRID:Addgene_186573 | see Materials and methods - Molecular Biology section |
| Recombinant DNA reagent | pMito-mCherry-FRB | *Wood et al., 2017* | RRID:Addgene_59352 | see Materials and methods - Molecular Biology section |
| Recombinant DNA reagent | σ1-mCherry | This paper | RRID:Addgene_186578 | see Materials and methods - Molecular Biology section |
| Recombinant DNA reagent | σ2-mCherry | *Willox and Royle, 2012* | RRID:Addgene_186579 | |
| Recombinant DNA reagent | σ3-mCherry | This paper | RRID:Addgene_186580 | see Materials and methods - Molecular Biology section |

## Molecular biology

The following plasmids were available from earlier work: FKBP-β2-GFP (WT and mutant versions), GFP-FKBP, FKBP-α-GFP, FKBP-β1-GFP, FKBP-β3-GFP, FKBP-epsin-GFP, pMito-mCherry-FRB, pMito-dCherry-FRB, CD8-dCherry-FRB, and σ2-mCherry (*Wood et al., 2017*; *Willox and Royle, 2012*). Rat Endophilin A1-RFP in pcDNA3.1 was a gift from L. Lagnado (University of Sussex). Plasmids for mCherry-Epsin-2 (#27673), mCherry-Fcho2 (#27686), Hip1r-tDimer-RFP (#27700), mCherry-Amphiphysin (#27692), and mCherry-Snx9 (#27678) (all mouse) were from Addgene.

To make the ER anchor FRB-mCherry-Sec61β, pAc-GFPC1-Sec61β (Addgene #15108) was inserted into pFRB-mCherry-C1 using BglII and EcoRI. The Golgi anchor FRB-mCherry-Giantin, was generated by excising Giantin (3131–3259) from pmScarlet-Giantin-C1 (Addgene #85050) with XhoI and BamHI and inserting into FRB-mCherry-C1. For the lysosome anchor Lamp1-mCherry-FRB, Lamp1 was amplified from LAMP1-mGFP (Addgene # 34831) and cloned in place of Tom70p in pMito-mCherry-FRB using EcoRI and BamHI.

The clathrin hook FKBP-AP180-GFP was made by amplifying residues 328–896 from rat AP180 (gift from E. Ungewickell) and ligating in place of β2 (616–951) in FKBP-β2-GFP using BamHI and AgeI.

Dynamin-1-mCherry and dynamin-1(K44A)-mCherry were made by replacing the GFP from WT Dyn1 pEGFP and K44A Dyn1 pEGFP (Addgene #34680 and #34681), respectively, with mCherry from pmCherry-N1 using AgeI and NotI. The plasmid to express mCherry-Drp1 K38A was made by site-directed mutagenesis of mCherry-Drp1 (Addgene #49152). Plasmids to express mCherry-Vps4a WT and mCherry-Vps4a E228Q were made replacing the GFP from GFP-Vps4wt and GFP-Vps4(EQmut), with mCherry from pmCherry-N1 using AgeI and BsrGI. Human AP1S1 and AP3S1 coding sequences with XhoI and BamHI ends, were synthesized as G-blocks (Integrated DNA Technologies) and cloned in place of σ2-mCherry to make σ1-mCherry and σ3-mCherry.

All plasmids developed as part of this work are available on Addgene.

## Cell biology

Wild-type HeLa cells (HPA/ECACC 93021013) or FCHO1/2 KO HeLa #64/1.E (*Umasankar et al., 2014*) were cultured in Dulbecco's modified Eagle medium (DMEM) with GlutaMAX (Thermo Fisher) supplemented with 10% fetal bovine serum (FBS), and 100 U ml⁻¹ penicillin/streptomycin. Dynamin triple knockout (DNM TKO) cells (*Park et al., 2013*), a kind gift from Pietro de Camilli (Yale School of Medicine) were cultured in DMEM supplemented with 10% FBS, 1% L-glutamine, 3.5% sodium bicarbonate and 1% penicillin/streptomycin. For conditional knockout, cells were treated for 48 hr with 3 µM 4-hydroxytamoxifen (Merck) and then kept in 300 nM until experimentation. All cells were kept at 37°C and 5% $CO_2$ and checked monthly for mycoplasma contamination.

HeLa or DNM TKO cells were transfected with GeneJuice (Merck) or Fugene (Promega), respectively, according to the manufacturer's instructions. Anchor and hook plasmids were transfected in a 1:2 (wt/wt) ratio. For clathrin heavy chain knockdown, HeLa cells were plated out and then transfected with GL2 (control, CGTACGCGGAATACTTCGA) or CHC siRNA (target sequence, TCCAATTC GAAGACCAATT) on days 2 and 4 using Lipofectamine2000 (*Motley et al., 2003*), with additional DNA transfection on day 4. Similarly for triple epsin knockdown, Hela cells were plated on cover slips and then transfected with a total of 600 pg of either a scrambled control siRNA oligo medium GC

content (Invitrogen) or a mix of three siRNA oligos HSS121071 (Epsin-1), HSS117872 (Epsin-2), and HSS147867 (Epsin-3) (Invitrogen) on days 2 and 3 (*Boucrot et al., 2012*), followed by additional DNA transfection on day 4. Cells were seeded onto cover slips and used for experiments on day 5.

Induction of CCPs was done by manual addition of rapamycin (Alfa Aesar) to a final concentration of 200 nM for 10 min to 30 min before fixation. Previously, we reported that no CCPs formed on mito-chondria using a related anchor, pMito-PAGFP-FRB (*Wood et al., 2017*). Using pMito-mCherry-FRB, we find MitoPits can be reproducibly induced in ~60% of cells expressing both constructs. We have verified that pits can also be formed using pMito-PAGFP-FRB, although the fraction of cells showing spots is lower, explaining our earlier report.

For transferrin uptake experiments, DNM TKO cells were serum starved for a total of 30 min in serum-free media. Then for rerouting, they were exposed to 200 nM rapamycin and 100 μg/ml Alexa Fluor 568- or 647-conjugated transferrin (Thermo Fisher) for the final 10 min of starvation. For HeLa cells, a dynamin inhibition step was added to the protocol where cells were treated with 30 μM Dynole 34–2, dynamin I and dynamin II inhibitor, or negative control Dynole 31–2 (Abcam, ab120474) starting from 15 min before rerouting and transferrin addition. For actin depolymerization, HeLa cells were treated with 1 μM Latrunculin B (Merck) or 0.02% vehicle (dimethyl sulfoxide, DMSO) for 25 min, with addition of 200 nM rapamycin for the final 10 min. All incubations were done in serum-free media at 37°C and 5% $CO_2$. Dose–response relationships for Dynole 34–2 and Latrunculin B were determined empirically as the lowest concentration to inhibit transferrin uptake or disrupt actin, respectively; without affecting mitochondrial morphology.

## Immunofluorescence

Cells were fixed in 4% formaldehyde, 4% sucrose in PEM buffer (80 mM piperazine-N,N′-bis(2-ethanesulfonic acid) (PIPES), 5 mM ethylene glycol-bis(β-aminoethyl ether)-N,N,N′,N′-tetraacetic acid (EGTA), and 2 mM $MgCl_2$, pH 6.8) for 10 min, washed three times with phosphate-buffered saline (PBS), then permeabilized for 10 min in 0.1% Triton X-100 in PBS. Cells were blocked in blocking solution (3% bovine serum albumin, 5% goat serum in PBS) for 60 min. Cells were then incubated for 60 min with primary antibodies diluted in blocking solution as follows: mouse anti-Pyruvate dehydrogenase E2/E3 (PDHE2/E3, ab110333, Abcam, 1 μg ml$^{-1}$); mouse anti-clathrin heavy chain (X22, 1:1000); rabbit anti-TOMM20-Alexa Fluor 647 (Abcam, ab209606, 0.5 μg ml$^{-1}$); acti-stain 555 phalloidin (Cytoskeleton Inc, PHDH1, 1:1000). Where secondary detection was required, cells were washed three times with PBS and incubated with anti-mouse IgG Alexa Fluor 568 (Thermo Fisher, A11031) or anti-mouse IgG Alexa Fluor 647 (Thermo Fisher, A21235) at 1:500 in blocking solution. After a final three washes with PBS, cover slips were then mounted using Mowiol. All steps were at room temperature.

## Microscopy

For live cell imaging, HeLa cells were grown in 4-well glass-bottom 3.5 cm dishes. Growth medium was exchanged for Liebovitz L-15 $CO_2$-independent medium (Gibco) before imaging. For some experiments, MitoTracker Deep Red FM (Thermo Fisher) was added at 1:15,000 to visualize the mitochondria.

Imaging was done using a Nikon CSU-W1 spinning disc confocal system with SoRa upgrade (Yokogawa), 60 × 1.4 NA oil-immersion objective (Nikon) with ×4 SoRa magnification and 95B Prime (Photometrics) camera was used with excitation by 405, 488, 561, or 638 nm lasers. Images were acquired with NIS-Elements software (Nikon). For CHC knockdown experiments, cells were imaged using an Ultraview Vox system (Perkin Elmer) with 100 × 1.4 NA oil objective and a Hamamatsu ORCA-R2 camera with excitation by 488, 561, or 640 nm lasers, operated by Volocity 6.0 software (Perkin Elmer).

To correlate light microscopy with EM, HeLa cells were plated onto gridded glass culture dishes (P35G-1.5-14-CGRD, MatTek Corporation, Ashland, MA, USA) at 30,000 cells per dish, transfected the following day, and imaged on a Nikon Ti-U widefield microscope with CoolSnap MYO camera (Photometrics) using NIS-Elements software. Location of each cell of interest was recorded using the coordi-nates on the grid at ×20 magnification with brightfield illumination. Cells were imaged live with a ×100 objective while rapamycin was added as described, and then cells were fixed in 3% glutaraldehyde, 0.5% formaldehyde in 0.05 M phosphate buffer, pH 7.4 for 1 hr and washed with phosphate buffer three times afterwards. Cells were stained with 1% osmium tetroxide, 1.5% potassium ferrocyanide for

1 hr, washed four times with distilled water for 5 min, stained with 1% tannic acid for 45 min, washed three times with distilled water for 5 min and stained with 1% uranyl acetate overnight at 4°C.

On day 6, cells were washed three times with distilled water for 5 min, dehydrated through ascending series of ethanol solutions (30%, 50%, 70%, 80%, 90%, 100 %, 10 min each) and infiltrated in medium epoxy resin (TAAB Laboratories Equipment Ltd., Aldermasteron, UK) at 2:1, 1:1, 1:2 ethanol to resin ratios and finally in full resin, each for 30 min. Fresh full resin was added, and a gelatin capsule was placed over each grid that contained the cell of interest. Resin was left to polymerize at 60°C for 72 hr.

On day 9, the cell of interest was located, and then trimmed down. Next, 80 nm serial sections were taken using a diamond knife and collected on Formvar coated copper hexagonal 100-mesh grids (EM Resolutions). Sections were poststained in 2% uranyl acetate for 2 min and in 3% Reynolds lead citrate (TAAB) for 2 min, with intermediate washes in distilled water.

Electron micrographs were taken on a JEOL 2100Plus transmission electron micrograph operating at 200 kV using Gatan OneView IS camera with GMS3.0 and TEMCenter software. Cells were imaged at low magnification (×100–400) to locate and then high resolution images were taken at ×25,000 magnification.

## Data analysis

To analyze the spatial organization of MitoPits, a line perpendicular to the mitochondrial axis was drawn through the MitoPit (situated on mitochondrial edges). To eliminate distortion, all images were registered prior to analysis using images of 200 nm fluorescent beads with NanoJ plugin (*Laine et al., 2019*). Intensity data for each channel as a function of distance were read into Igor Pro and a 1D Gaussian fitting procedure was used to locate the peak for each channel, offset to the peak for FKBP-β2-GFP, and generate an ensemble average.

Colocalization analysis of spots, formed by rerouting of FKBP-β2-GFP to dark MitoTrap, with another (mCherry-tagged) protein was done using the ComDet Plugin v0.5.5 in Fiji (*Katrukha, 2022*). The maximum distance between spots of FKBP-β2-GFP and spots in the other protein's channel to be accepted as colocalization was selected to be 2 pixels, and spot size was 3 pixels (corresponding to 135 nm). This method likely underestimates colocalization where the density of spots in one channel is higher than in another, which is the case for clathrin immunostaining due to the large number of endogenous non-MitoPit clathrin spots. For CHC knockdown experiments, total spots in a cell were counted in Fiji using 'Analyze Particles'. Briefly, spots were isolated by applying manual threshold to images in the FKBP-β2-GFP channel, and analyzed particles with limits of 0.03–1.5 µm in size and 0.4–1.0 circularity, counting the number of spots for each cell.

For MitoPit location analysis, multiple square ROIs (150 × 150 pixels) that contained a single distinguishable mitochondrion were selected. Spots that were classified as free, or on the edges, or at nodes (branchpoints and endpoints) were counted using Cell Analyzer in Fiji (*Schindelin et al., 2012*). To calculate the surface area at edges and endpoints, Z-stacks (0.5 µm step size) of HeLa cells stably expressing MitoTrap were analyzed using MitoGraph (*Viana et al., 2015*). Using R, the MitoGraph outputs were processed and the surface area of edges was calculated using the average width of mitochondria ($2r$) and the total length of edges ($l$) using $l2\pi r$. The surface area of ends was calculated assuming each was a hemisphere ($2\pi r^2$) and the number was derived from the fraction of nodes in the MitoGraph that were designated free ends.

Spot detection for efficiency measurements and free spot analysis was done either using NIS-Elements Advanced Research analysis software or an equivalent script in Fiji. The total number of spots for each cell was counted using the 'Spot Detection Binary' function (3 pixels, corresponding to 135 nm) using the FKBP-β2-GFP channel with a manual threshold. This measurement was normalized to the cell area to give the spot density per unit area. Mitochondria were recognized by the 'Homogeneous Area Detection Binary' function using the mitochondrial matrix channel with a manual threshold. The coincidence of detected spots with this segmented area represented the MitoPits, while those spots outside it were designated free spots. The equivalent script for Fiji is available (see below). To quantify the size and abundance of hot-wired endocytic vesicles upon dynamin inhibition with Dynole 34–2, custom-written code for Fiji and Igor Pro 9 was used. Briefly, a mask for each cell was made via thresholding using the IsoData algorithm. Then, these masks were analyzed using 'Analyze Particles' function in Fiji, with limits of 0–1 µm in size and 0.3–1.0 in circularity, counting the number of spots for

each cell and measuring the area of the spots. Analysis was done with the experimenter blind to the conditions of the experiment. Figures were made with FIJI and Igor Pro, and assembled using Adobe Illustrator. Null hypothesis statistical tests were done as described in the figure legends.

### Data and software availability

Data and code to analyze MitoPit profiles from light and electron microscopy images are available at https://github.com/quantixed/p057p034, (copy archived at swh:1:rev:d01fe5cb80f9741bc-83527def95790ffb049aba1) (*Royle, 2022*). Raw data associated with plots in the paper and code to reproduce plots is also available in this repo.

## Acknowledgements

We thank Claire Mitchell and Laura Cooper from the Computing and Advanced Microscopy Unit (CAMDU) for their help and support; Corinne Smith, Antoine Allard, Darius Koester, and members of the Royle lab for useful discussions; Saskia Bakker for help with electon microscopy. CK was supported by University of Warwick, and the EPSRC and BBSRC Centre for Doctoral Training in Synthetic Biology (grant EP/L016494/1). MS was supported by a grant from UKRI-BBSRC (BB/V003062/1).

## Additional information

### Funding

| Funder | Grant reference number | Author |
|---|---|---|
| UK Research and Innovation | EP/L016494/1 | Cansu Küey |
| UK Research and Innovation | BB/V003062/1 | Stephen J Royle<br>Méghane Sittewelle |

The funders had no role in study design, data collection, and interpretation, or the decision to submit the work for publication.

### Author contributions

Cansu Küey, Formal analysis, Investigation, Writing - original draft; Méghane Sittewelle, Software, Investigation; Gabrielle Larocque, Investigation; Miguel Hernández-González, Conceptualization; Stephen J Royle, Software, Formal analysis, Supervision, Visualization, Writing - original draft, Writing - review and editing

### Author ORCIDs

Cansu Küey ⓘ http://orcid.org/0000-0002-7992-3523
Méghane Sittewelle ⓘ http://orcid.org/0000-0002-9383-6653
Gabrielle Larocque ⓘ http://orcid.org/0000-0001-8295-9378
Miguel Hernández-González ⓘ http://orcid.org/0000-0002-5562-7076
Stephen J Royle ⓘ http://orcid.org/0000-0001-8927-6967

### Decision letter and Author response

Decision letter https://doi.org/10.7554/eLife.78929.sa1
Author response https://doi.org/10.7554/eLife.78929.sa2

## Additional files

### Supplementary files
• MDAR checklist

### Data availability

Data and code associated with this study are available at https://github.com/quantixed/p057p034, (copy archived at swh:1:rev:d01fe5cb80f9741bc83527def95790ffb049aba1).

The following dataset was generated:

| Author(s) | Year | Dataset title | Dataset URL | Database and Identifier |
| --- | --- | --- | --- | --- |
| Royle SJ, Sittewelle M | 2022 | p057p034 | https://github.com/quantixed/p057p034 | GitHub, 3cd66d3 |

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
