## [Editor Report]

This paper reports a striking finding, which should be of interest to cell biologists and biophysicists. The authors use an innovative approach to recruit clathrin to mitochondrial membranes, and observe the budding and fission of clathrin-coated vesicles. The study leads to a much clearer view of how the clathrin lattice functions in endocytosis.

---

## [Decision Letter]

**Decision letter after peer review:**

Thank you for submitting your article "Recruitment of clathrin to intracellular membranes is sufficient for vesicle formation" for consideration by *eLife*. Your article has been reviewed by 3 peer reviewers, one of whom is a member of our Board of Reviewing Editors, and the evaluation has been overseen by Vivek Malhotra as the Senior Editor. The reviewers have opted to remain anonymous.

All three reviewers thought your article was interesting and innovative, with high-quality data. However, there was also a general consensus that some of your conclusions need to be tempered, and that some additional papers need to be cited.

Essential revisions:

1) The statement that "our work suggests that from the moment of recruitment onwards, clathrin is sufficient to form a vesicle" needs to be toned down. First, the mitochondrial membranes are already highly curved; your own studies show that the situation is different with a flat membrane. Second, the clathrin "hook" is a bulky disordered protein of the sort that can deform membranes, and the fact that this doesn't happen unless clathrin is added is likely due to the fact that the proteins don't cluster on their own. Thus, it is premature to conclude that all the other proteins associated with clathrin-coated pits are modulators rather than drivers of membrane bending. The reviewers' comments below give clear indications of how this part of the discussion needs to be rewritten.

2) A particularly pertinent paper is the recent preprint in bioRxiv from the Drubin lab. These authors came to somewhat different conclusions from yours, using cells grown on substrates that cause the ventral plasma membrane to bend. This discrepancy needs to be discussed. Other citations are also suggested; see the reviewers' comments.

3. Some of the biophysical calculations do not take the full situation into account and need to be revised; see below for details.

*Reviewer #1 (Recommendations for the authors):*

As detailed below, I had a couple of comments about membrane fluidity and molecular crowding, which would be good to clarify.

I do have a few comments and questions for the authors to consider.

1. The authors discuss the flatness of the plasma membrane compared with intracellular membranes like the TGN and endosomes, and they highlight their very interesting observation that the MitoPits most often form on curved parts of mitochondria. It might be pertinent to cite a 1983 paper from Zena Werb's lab on unroofed cells, showing that clathrin-coated buds on intracellular membranes form from curved edges, either tubules or the rims of flat cisternae (see PubMed 6415067).

2. The authors discuss the fluidity of the mitochondrial membrane vs. the plasma membrane (page 8). I'm not a biophysicist, but I would have thought it would be more difficult to bend a double membrane than a single one. Perhaps the authors could comment on this.

3. The authors argue against a curvature role for the proteins with clathrin-binding domains that they use as hooks because they don't get MitoPits forming when they mutate the clathrin binding site. But I'm not sure they can completely rule out molecular crowding as a contributing factor. I take the point that clathrin is required, but if clathrin were acting as a crosslinker, bringing together multiple copies of their hooks, all of which contain disordered regions, couldn't both mechanisms be involved?

*Reviewer #2 (Recommendations for the authors):*

I have a few comments which could be addressed with new text, analysis, or experiments. Specifically, a more thorough discussion of past work would be helpful for the reader. Finally, the screen of co-localized endocytic proteins was somewhat limited and lends the reader to wonder if other key factors, particularly eps15/R, might still be involved. It is, however, quite a beautiful study.

Comments for Authors to address:

1. Figure 2. The scale and degree of curvature encountered at the mitochondria by the assembling clathrin lattice is an important consideration in this work. To compare the current work to past work where clathrin assembly was enhanced by nanofabricated pre-curved surfaces, it would be helpful to generally quantitate the degree of curvature observed at the edges, branches, or ends of the mitochondria. The curvatures studied in past work (Zhao et al. NatNano2017, Cali et al. BioRxiv 2021) were fairly extreme and effects similar to those seen here. Can the authors please expand the discussion or provide some additional analysis of the types of mitochondria curvatures encountered in their thin section TEM images of these cells? Likewise, a discussion of how this work compares to a preprint showing that curved surfaces do not need clathrin would be helpful (Cali and Drubin bioRxiv 2021: Induced membrane curvature bypasses clathrin's essential endocytic function).

2. Figure 3. If only 60% of the FKBP2-Beta2-GFP spots have clathrin staining after rapamycin treatment, what are the other green spots? Do they have a very low (dim) level of clathrin on them? Are they something else? Please discuss.

3. Figure 6. The percent of Epsin/GFP-B2 hook co-localization is higher than clathrin/GFP-B2 hook co-localization. Can the authors explain this effect? Specifically, If clathrin is needed for epsin colocalization to mitopits, and only 60% of the mitopits have clathrin, why would 80% of the mitopits have a strong epsin signal? If the experiment with clathrin colocalization were done with CLC-mCherry and a dark-mitotrap, would the authors see a higher colocalization than the CHC antibody staining? Please address this issue in the text or with a new analysis.

4. "Our work suggests that from the moment of recruitment on-wards, clathrin is sufficient to form a vesicle."

I believe the statement should be more nuanced. The authors show that clathrin alone can form at mitochondrial membranes. However, the fact that it occurs much more robustly at pre-curved membranes implies that other factors are likely needed at flat membranes. Likewise, the efficiency of the formation of the mitopits is not measured in this paper. Is it fundamentally slower or less efficient than what naturally occurs at the plasma membrane? These are important issues to consider. Please discuss.

5. "the clathrin-only mechanism we describe suggests that none of these proteins are 'mediators' of vesicle formation, and instead they may act as 'modulators'"…..

See the above comment. It is possible that other factors are needed (mediators) at the plasma membrane when the lattice grows on flat pieces of the plasma membrane. I think this distinction (mediator vs. modulator) in the context of CME is not necessarily addressed by this paper.

6. "previously, we reported that no clathrin-coated pits formed on mitochondria using a related anchor, pMito-PAGFP-FRB (Wood et al., 2017). Using pMito-mCherry-FRB, we find MitoPits can be reproducibly induced in ˘60 % of cells expressing both constructs. We have verified that pits can also be formed using pMito-PAGFP-FRB, although the fraction of cells showing spots is lower, explaining our earlier report.'

This is an interesting finding. Can the authors explain these results? Why do only 60% of the cells show the effect, and why was assembly not seen with the other hook tethers? Please discuss.

7. What was the logic for choosing the accessory proteins to screen for co-localization with mitopits (Figure 6 and S5)? It would be interesting to test if Esp15/R is co-localized to mitopits. Eps15 binds espin and was pulled down in the mass-spec Beta2 pulldown experiments (Schmid et al. PLOSBIO 2006). The other major protein found in the mass-spec data was intersectin. In a similar vein, does Eps15 knockdown or Eps15 domain expression prevent mitopit formation? If Eps15 is strongly localized and has a functional impact on mitopit formation, these data would adjust the core findings of the paper and add an additional component along with clathrin that is necessary for mitopit formation. In the least, the authors should discuss this possibility.

8. I would recommend shifting to a magenta/green or similar color scheme for color-blind readers.

9. The paper could use some additional citations for consideration and discussion.

Hassinger et al. PNAS 2017

Heuser JCB 1989

Sochacki et al. Dev Cell 2021

Lin et al. JCB 1991

*Reviewer #3 (Recommendations for the authors):*

A thorough rewriting of the discussion is necessary to remove unnecessary speculation and remove logical flaws in the authors' interpretation of the data:

1. The authors must acknowledge that their findings are exclusive to pre-curved membranes. The title should be amended to include that idea. "Recruitment of clathrin to highly curved intracellular membranes is sufficient for vesicle formation"

2. The authors must cite and discuss the Cail and Drubin paper, noting that adapters are also capable of functioning without clathrin.

3. The authors must either acknowledge that molecular crowding cannot be eliminated as a possible explanation for their findings or provide a detailed stoichiometric analysis as described above. Note this analysis must account for all of the proteins that reside under the clathrin coat, including epsin, fcho, etc. In this way, it may prove challenging to come to a definitive conclusion.

4. The authors should remove statements suggesting that all of clathrin's adaptor proteins are only "modulators" rather than "drivers" of membrane remodeling. This conclusion is clearly incorrect based on the Cail and Drubin work.

---

## [Author Response]

Essential revisions:1) The statement that "our work suggests that from the moment of recruitment onwards, clathrin is sufficient to form a vesicle" needs to be toned down. First, the mitochondrial membranes are already highly curved; your own studies show that the situation is different with a flat membrane. Second, the clathrin "hook" is a bulky disordered protein of the sort that can deform membranes, and the fact that this doesn't happen unless clathrin is added is likely due to the fact that the proteins don't cluster on their own. Thus, it is premature to conclude that all the other proteins associated with clathrin-coated pits are modulators rather than drivers of membrane bending. The reviewers' comments below give clear indications of how this part of the discussion needs to be rewritten.

We have reworked the paper to tone down the text and temper our message.

On the second point, we have addressed this issue by specifically describing that disordered proteins can contribute to membrane bending after being clustered by clathrin. We have also edited the text to remove instances of "clathrin only" and make it clear that the hook and clathrin are required.

On the first, the curvature preference for MitoPit formation is an interesting aspect of our work, but note that an analogous system can be used to make CCVs at the plasma membrane (Wood et al., 2017 PMID: 28954824); so curvature is clearly not a pre-requisite.

We respond in detail on these points below.

2) A particularly pertinent paper is the recent preprint in bioRxiv from the Drubin lab. These authors came to somewhat different conclusions from yours, using cells grown on substrates that cause the ventral plasma membrane to bend. This discrepancy needs to be discussed. Other citations are also suggested; see the reviewers' comments.

We are sorry to have not discussed this work in the original manuscript. We now cover their paper in the Discussion and below we respond in detail. There are important differences between the two studies, for example the scale of curvature used in Cail et al. is quite different to ours, and so a comparison between them is complicated.

Note that, while we were revising our paper, a Spotlight article was published on the work of Cail et al. (Stachowiak 2022 PMID: 35704021) which compares our preprinted work with theirs; so others in the field have already had their say!

3. Some of the biophysical calculations do not take the full situation into account and need to be revised; see below for details.

Please see our full response below. In brief, the criticisms raised by the referee do not substantially alter the calculation that was in the paper. Briefly, in the opinion of Reviewer #3 the value we used for rigidity of the mitochondrial outer membrane was too low. We now make it more clear that the value used in the calculation actually comes from a real world measurement. The Referee's other concern was about the energy barrier to curvature initiation, but we show below that this is negligible.

Please note, that our calculation is not a formal attempt to model the process of MitoPit formation, which should consider other aspects. Our goal is simply to provide an estimate for the reader who is probably wondering whether what we are proposing is energetically feasible.

Reviewer #1 (Recommendations for the authors):As detailed below, I had a couple of comments about membrane fluidity and molecular crowding, which would be good to clarify.I do have a few comments and questions for the authors to consider.1. The authors discuss the flatness of the plasma membrane compared with intracellular membranes like the TGN and endosomes, and they highlight their very interesting observation that the MitoPits most often form on curved parts of mitochondria. It might be pertinent to cite a 1983 paper from Zena Werb's lab on unroofed cells, showing that clathrin-coated buds on intracellular membranes form from curved edges, either tubules or the rims of flat cisternae (see PubMed 6415067).

Thank you for this suggestion. We have included a citation to this paper when discussing MitoPit formation at pre-curved sites.

“This preference for pre-curved sites echoes classic work showing clathrin-coated pits forming at the curved edges of intracellular membranes (Aggeler et al., 1983), while a very recent study suggests that clathrin prefers pre-curved surfaces (Zeno et al., 2021).”

2. The authors discuss the fluidity of the mitochondrial membrane vs. the plasma membrane (page 8). I'm not a biophysicist, but I would have thought it would be more difficult to bend a double membrane than a single one. Perhaps the authors could comment on this.

In our rough calculation, we factored in the difficulty of bending a double membrane *versus* a single one. This was done by simply taking two times the energy required to form a vesicle, i.e. **2 ×** (2*κ/r*^2^ + *γ*). It is possible that coupling between the two membranes decreases (or even increases) this requirement, but taking two times the energy seemed the most cautious approach. We have edited this part of the paper in light of comments from Reviewer #3 and now the double membrane issue is better signposted and explicitly mentioned as an unknown parameter which may affect the result.

3. The authors argue against a curvature role for the proteins with clathrin-binding domains that they use as hooks because they don't get MitoPits forming when they mutate the clathrin binding site. But I'm not sure they can completely rule out molecular crowding as a contributing factor. I take the point that clathrin is required, but if clathrin were acting as a crosslinker, bringing together multiple copies of their hooks, all of which contain disordered regions, couldn't both mechanisms be involved?

This is an excellent point which was also raised by Reviewer #3. It is difficult to separate the contributions of clathrin and molecular crowding to curvature generation; since clathrin *and* hook are both needed to make MitoPits. Our experiments rule out the possibility that hooks drive curvature *alone*, but the possibility that molecular crowding – organised by clathrin – contributes to curvature remains. We have edited this part of the discussion so that it now reads:

“The clathrin hooks that we use all contain intrinsically disordered regions and such proteins have been shown to deform membranes via a phase separation mechanism (Busch et al., 2015; Yuan et al., 2021). Importantly, a clathrin-binding deficient β2 hook which differs from the wild-type by only a few residues was unable to support MitoPit formation, arguing against a contribution from the hook alone via this mechanism. It is tempting therefore to conclude that curvature generation in the context of MitoPits is by clathrin alone. However, it is possible that the disordered region of the hook contributes to curvature generation in a manner that is organized by clathrin, i.e. that the clathrin lattice could spatially constrain the hook and, when concentrated, the disordered regions may contribute to curvature. Separating these possibilities is difficult since both clathrin and hook are essential to make MitoPits. Whatever the mechanism, it seems that enough force is generated by our synthetic system to deform both the inner and outer mitochondrial membranes and even to pinch off the MitoPits.”

We have also edited the calculation to include this concession (details given below in response to Reviewer #3).

Reviewer #2 (Recommendations for the authors):I have a few comments which could be addressed with new text, analysis, or experiments. Specifically, a more thorough discussion of past work would be helpful for the reader. Finally, the screen of co-localized endocytic proteins was somewhat limited and lends the reader to wonder if other key factors, particularly eps15/R, might still be involved. It is, however, quite a beautiful study.Comments for Authors to address:1. Figure 2. The scale and degree of curvature encountered at the mitochondria by the assembling clathrin lattice is an important consideration in this work. To compare the current work to past work where clathrin assembly was enhanced by nanofabricated pre-curved surfaces, it would be helpful to generally quantitate the degree of curvature observed at the edges, branches, or ends of the mitochondria. The curvatures studied in past work (Zhao et al. NatNano2017, Cali et al. BioRxiv 2021) were fairly extreme and effects similar to those seen here. Can the authors please expand the discussion or provide some additional analysis of the types of mitochondria curvatures encountered in their thin section TEM images of these cells? Likewise, a discussion of how this work compares to a preprint showing that curved surfaces do not need clathrin would be helpful (Cali and Drubin bioRxiv 2021: Induced membrane curvature bypasses clathrin's essential endocytic function).

The work of Cail et al. (now published in J Cell Biol PMID: 35532382) is very interesting indeed. The Reviewer is correct that the nanofabrication-imposed curvature in these studies is fairly extreme. They used ridges with widths of 75 nm to 500 nm, and see significant enrichment of adaptors at 200 nm and narrower, i.e. on the scale of a clathrin-coated pit. The width of a mitochondrion is ~500 nm, so the MitoPits that form on the edges of mitochondria in our study are not in the range where Cail et al. report significant enrichment.

We have measured the curvature in our thin section TEM images, as suggested. With the limitation that these are 2D views that could exaggerate curvature, and assuming that curvature in the unseen dimension has a fixed radius of 250 nm, the results allow a coarse comparison with the nano-fabrication work (**Reviewer Figure 1**).

**Author response image 1. sa2fig1:** Estimated curvature at mitochondria bearing MitoPits. Plot to show Gaussian curvature (K, product of two curvatures) versus total curvature (J, sum of two curvatures). Crosses show our measurements compared to theoretical values (red circles). The range of curvatures that show enrichment in the work of Cail et al. 2022 (Figure 1D) are indicated (blue). Note that our measured values lie on a line due to the substitution of 1/250 for curvature in the 3rd dimension.

In response to this comment and that of Reviewer #3 we now discuss the work of Cail et al. in more detail in our paper.

The curvature preference for MitoPit formation is an interesting aspect of our work, but note that an analogous system can be used to make CCVs at the plasma membrane (Wood et al., 2017 PMID: 28954824). So curvature is clearly not a pre-requisite.

2. Figure 3. If only 60% of the FKBP2-Beta2-GFP spots have clathrin staining after rapamycin treatment, what are the other green spots? Do they have a very low (dim) level of clathrin on them? Are they something else? Please discuss.

We used an object-based colocalization method which works on the basis of spot detection. The level of colocalisation is therefore affected by the signal-to-noise level used for detection and by the density of spots. When we crank the clathrin channel we can see that many of these apparently FKBP-beta2-GFP-only spots do have a low level of clathrin on them. If we decrease the threshold for detection, the density of clathrin spots (MitoPits plus endogenous CCPs) tend to merge and then not get detected as spots due to their size. The value of 60% shouldn’t be fixated on as it is an underestimate. However, this approach is still useful to look at changes within an experiment, for example between wt and mutant hooks. We have added a note to this effect in the legend of Figure 3 and in the Materials and methods section.

Having said all this, even with the clathrin signal cranked, there are some FKBP-beta2-GFP spots that are clathrin-negative. We assume that these are MPDVs that have uncoated.

3. Figure 6. The percent of Epsin/GFP-B2 hook co-localization is higher than clathrin/GFP-B2 hook co-localization. Can the authors explain this effect? Specifically, If clathrin is needed for epsin colocalization to mitopits, and only 60% of the mitopits have clathrin, why would 80% of the mitopits have a strong epsin signal? If the experiment with clathrin colocalization were done with CLC-mCherry and a dark-mitotrap, would the authors see a higher colocalization than the CHC antibody staining? Please address this issue in the text or with a new analysis.

In continuation of our response above, we would be cautious about drawing connections between the 80% colocalization in these experiments with 60% in the other. For example, the density of epsin spots is lower than that of clathrin and so the apparent colocalisation is higher. We have imaged mCherry-LCa with FKBP-beta2-GFP and dark MitoTrap to see if the colocalisation improves and we found it similar to our results using CHC staining to detect clathrin.

We have explored alternative methods for these colocalisation analyses, with a view to improving these parts of the paper, but found that they have other limitations. So we have opted to stay with the current approach.

It is possible that the higher apparent colocalisation with epsin compared to clathrin is because epsin stays associated with MPDVs after they have lost clathrin. We wrote in the paper that we think that the higher colocalisation of Epsin2 vs Fcho2 at MitoPits may be due to Epsin's ability to bind beta2 and clathrin. The beta2 association may explain the difference the reviewer is asking about. In a test of this idea, we expressed FKBP-beta2-GFP with Y815A/Y888V mutation, which is unable to bind accessory proteins (Edeling et al., 2006 PMID: 16516836). Unfortunately, this mutation also affects the association with clathrin, so the total number of MitoPits is decreased. However, within this population, there are fewer Epsin-positive MitoPits (**Reviewer Figure 2**).

**Author response image 2. sa2fig2:** mCherry-Epsin2 recruitment to MitoPits. Lower colocalisation is seen using a Y815A/Y888A mutant form of FKBP-beta2-GFP to form MitoPits compared to wild type.

4. "Our work suggests that from the moment of recruitment on-wards, clathrin is sufficient to form a vesicle."I believe the statement should be more nuanced. The authors show that clathrin alone can form at mitochondrial membranes. However, the fact that it occurs much more robustly at pre-curved membranes implies that other factors are likely needed at flat membranes. Likewise, the efficiency of the formation of the mitopits is not measured in this paper. Is it fundamentally slower or less efficient than what naturally occurs at the plasma membrane? These are important issues to consider. Please discuss.5. "the clathrin-only mechanism we describe suggests that none of these proteins are 'mediators' of vesicle formation, and instead they may act as 'modulators'"…..See the above comment. It is possible that other factors are needed (mediators) at the plasma membrane when the lattice grows on flat pieces of the plasma membrane. I think this distinction (mediator vs. modulator) in the context of CME is not necessarily addressed by this paper.

We thank the reviewer for their input (points 4 and 5). They are absolutely correct that more nuance was required to this and similar statements in the paper. By “clathrin alone” and “clathrin-only” we meant “no additional factors”, but this description overlooks the potential contribution of (i) membrane curvature and (ii) the hook used to recruit clathrin. A second issue is how our findings relate to CME at the plasma membrane.

We have removed all mentions of “clathin alone” and “clathrin-only” and refer only to a “clathrin-centric” mechanism. We really like the sentence highlighted by the Reviewer: “Our work suggests that from the moment of recruitment onwards, clathrin is sufficient to form a vesicle.” And couldn’t bear to change it. We have added nuance to this paragraph by stating two sentences later:

“It is important to note that in our system, we trigger the recruitment of clathrin to a pre-curved intracellular surface using a hook that may be active, and we are blind to potential mediators (cargo, lipids and other proteins) acting earlier. However, since clathrin recruitment defines the first stage of CCV formation – initiation – the proposed mechanism accounts for almost the entire pathway.”

The only place where we talk about clathrin-only is now the part where we specifically calculate whether clathrin alone could be responsible for vesicle formation. This section has also been significantly edited in response to the other Reviewers’ comments.

To tackle the second issue we have separated out the discussion of implications for CME from our calculations of MPDV budding and now have a distinct paragraph on this issue. Our previous discussion over-emphasised the “clathrin-centric” view and we realised it would be helpful to collate the differences between the two here. The paragraph continues:

“Given the differences between intracellular CCV formation and CME, the distinction between mediators and modulators at the plasma membrane may differ slightly.”

The Reviewer asks whether MitoPit formation is slower or less efficient than that which naturally occurs at the plasma membrane. We haven’t done a direct comparison to know for sure. Such a comparison is difficult perhaps the best way to do this would be *indirectly* with a lifetime analysis in TIRF of endogenous CME compared to hot-wired endocytosis and compare that data with MitoPit formation via confocal imaging and spot tracking.

6. "previously, we reported that no clathrin-coated pits formed on mitochondria using a related anchor, pMito-PAGFP-FRB (Wood et al., 2017). Using pMito-mCherry-FRB, we find MitoPits can be reproducibly induced in ˘60 % of cells expressing both constructs. We have verified that pits can also be formed using pMito-PAGFP-FRB, although the fraction of cells showing spots is lower, explaining our earlier report.'This is an interesting finding. Can the authors explain these results? Why do only 60% of the cells show the effect, and why was assembly not seen with the other hook tethers? Please discuss.

We don’t fully understand why the efficiency is ~60%. Different hooks displayed a similar efficiency (beta1, AP180, epsin). It appears to be some combination of: (1) levels of transient expression of the FKBP and FRB containing proteins (our knocksideways experiments with knock-in of FKBP at the endogenous locus are higher efficiency); (2) cell state possibly affecting rapamycin accessibility to the cell (MitoPits can be triggered in mitotic cells, but the efficiency is ~30%); (3) experimenter-to-experimenter variability. The efficiency is sufficient to do our experiments and therefore we have not made a huge effort to increase it.

The anchor effect is strange. We have previously observed that the spacing of FRB from the mitochondria makes a difference for knocksideways experiments and although pMito-PAGFP-FRB is not very different from pMito-mCherry-FRB or pMito-dCherry-FRB, it is possible that this accounts for the difference.

7. What was the logic for choosing the accessory proteins to screen for co-localization with mitopits (Figure 6 and S5)? It would be interesting to test if Esp15/R is co-localized to mitopits. Eps15 binds espin and was pulled down in the mass-spec Beta2 pulldown experiments (Schmid et al. PLOSBIO 2006). The other major protein found in the mass-spec data was intersectin. In a similar vein, does Eps15 knockdown or Eps15 domain expression prevent mitopit formation? If Eps15 is strongly localized and has a functional impact on mitopit formation, these data would adjust the core findings of the paper and add an additional component along with clathrin that is necessary for mitopit formation. In the least, the authors should discuss this possibility.

The initial motivation was to look for proteins that are reported to act upstream of dynamin in CME, e.g. amphiphysin, endophilin. We expanded to look at other accessory proteins, but it is clearly not a comprehensive survey. The question about Eps15 is interesting and we have now looked at this (**Reviewer Figure 3**). In short, we can see recruitment of endogenous Eps15 to MitoPits. If we express a dominant negative mutant known to inhibit CME (EH29, Benmerah et al. 1999 PMID: 10194409) which mislocalises endogenous Eps15, we still see the formation of MitoPits and MPDVs, which suggests that Eps15 is not essential for MitoPit formation. This fits with the general pattern that some accessory proteins, which can bind beta2 and/or clathrin, are forced to arrive at the MitoPIt even though they are not required; either because it is too late in pit formation, or that they cannot function at the mitochondrial outer membrane.

We have not included these new experiments in the paper, because we found that overexpression of WT Eps15-mCherry interferes with normal MitoPit formation (see large MitoPits, large endogenous Eps15 puncta and mitochondrial aggregates in **Reviewer Figure 3**). It's not clear to us if this artefact has any functional meaning. From these experiments though we can see that if we displace Eps15 with the dominant-negative mutant there is no effect on MPDV formation.

**Author response image 3. sa2fig3:** Eps15 recruitment to MitoPits. (A) MitoPit formation in the presence of Eps15-mCherry or mutant EH29. Endogenous Eps15 (detected by IF) is recruited to MitoPits. (B) The fraction of Free Spots (MPDVs) is unaffected by co-expression of EH29. (C) Expression of EH29 mislocalises Eps15. Note that overexpression of Eps15-mCherry causes aggregation of large Eps15 structures and MitoPits.

8. I would recommend shifting to a magenta/green or similar color scheme for color-blind readers.

Our strategy for making accessible figures is to use red/green[/blue] for merges *and* to show the separate channels in grayscale for colour blind readers to assess the channels independently. Prompted by this comment we used a macro in Fiji to determine accessibility of all of our figures; we found several that were not colour blind-safe. We have corrected them as follows:

Figure 2: MitoPit locations (A) now shown in magenta/green only. Bar chart (B) was distinguishable in the original colour scheme, but labels have been added.Figure 4: The purple/blue scheme for MPDV profiles (C) has been changed to red/blueVideo 1: shifted to magenta/greenVideo 2: shifted to magenta/green

The Waffle Plots shown in Figures3 and 6 are red/green but have been labelled and are distinguishable by colour blind readers.

9. The paper could use some additional citations for consideration and discussion.Hassinger et al. PNAS 2017Heuser JCB 1989Sochacki et al. Dev Cell 2021Lin et al. JCB 1991

We thank the Reviewer for these suggestions. Apologies for our previous omission of the Sochacki et al. 2021 paper which was particularly relevant for us to discuss. All four papers were pertinent and have been cited in the new part of the Discussion which compares MitoPit formation with CCP formation at the plasma membrane.

Reviewer #3 (Recommendations for the authors):A thorough rewriting of the discussion is necessary to remove unnecessary speculation and remove logical flaws in the authors' interpretation of the data:1. The authors must acknowledge that their findings are exclusive to pre-curved membranes. The title should be amended to include that idea. "Recruitment of clathrin to highly curved intracellular membranes is sufficient for vesicle formation"

The preference for MitoPit formation at precurved membranes is mentioned in the abstract. We do not wish to change the title of our paper. The sites we observed MitoPit formation on cannot really be described as *highly* curved. Plus MitoPits can form on edges with lower curvatures and branchpoints with a more complicated geometry.

2. The authors must cite and discuss the Cail and Drubin paper, noting that adapters are also capable of functioning without clathrin.

We now discuss this work in detail (see comments above and to Reviewer #2). Note that, while we were revising the paper, a Spotlight article has been published on the work of Cail et al. (Stachowiak 2022 PMID: 35704021) which compares our preprinted work with theirs; so others in the field have also already had their say!

3. The authors must either acknowledge that molecular crowding cannot be eliminated as a possible explanation for their findings or provide a detailed stoichiometric analysis as described above. Note this analysis must account for all of the proteins that reside under the clathrin coat, including epsin, fcho, etc. In this way, it may prove challenging to come to a definitive conclusion.

Please see our response above.

4. The authors should remove statements suggesting that all of clathrin's adaptor proteins are only "modulators" rather than "drivers" of membrane remodeling. This conclusion is clearly incorrect based on the Cail and Drubin work.

There were no statements in our paper of that kind. We describe in the introduction that cargo, adaptor, clathrin (± pinchase) are required to form a vesicle and question whether accessory proteins (not including adaptor proteins) are modulators rather than mediators of CCV formation.